# BMP signaling promotes zebrafish heart regeneration via alleviation of replication stress

Mohankrishna Dalvoy Vasudevarao[1,11], Denise Posadas Pena[1,11], Michaela Ihle[2], Chiara Bongiovanni[3,4], Pallab Maity [5], Dominik Geissler [1], Hossein Falah Mohammadi[1], Melanie Rall-Scharpf [2], Julian Niemann[6], Mathilda T. M. Mommersteeg [7], Simone Redaelli[8], Kathrin Happ[1], Chi-Chung Wu[1], Arica Beisaw [9], Karin Scharffetter-Kochanek [5], Gabriele D'Uva[3,4], Mona Malek Mohammadi [10], Lisa Wiesmüller [2], Hartmut Geiger [6] & Gilbert Weidinger [1]✉

In contrast to mammals, adult zebrafish achieve complete heart regeneration via proliferation of cardiomyocytes. Surprisingly, we found that regenerating cardiomyocytes experience DNA replication stress, which represents one reason for declining tissue regeneration during aging in mammals. Pharmacological inhibition of ATM and ATR kinases revealed that DNA damage response signaling is essential for zebrafish heart regeneration. Manipulation of Bone Morphogenetic Protein (BMP)-Smad signaling using transgenics and mutants showed that BMP signaling alleviates cardiomyocyte replication stress. BMP signaling also rescues neonatal mouse cardiomyocytes, human fibroblasts and human hematopoietic stem and progenitor cells (HSPCs) from replication stress. DNA fiber spreading assays indicate that BMP signaling facilitates re-start of replication forks after replication stress-induced stalling. Our results identify the ability to overcome replication stress as key factor for the elevated zebrafish heart regeneration capacity and reveal a conserved role for BMP signaling in promotion of stress-free DNA replication.

Myocardial infarction is a leading cause of death, since the human heart, like that of most adult mammals, lacks the ability to replace cardiomyocytes (CMs) lost to injury. In contrast, adult zebrafish and a few other highly regenerative animals can repair heart injuries via dedifferentiation and proliferation of spared, mature CMs located at the wound border[1–6]. Zebrafish CM regeneration is remarkably efficient and complete; in response to cryoinjuries that kill 1/3 of the ventricular CMs, pre-injury CM numbers are restored within 30 days[7]. Many signaling pathways have been found to promote zebrafish CM dedifferentiation and cell cycling[8–10]. We have previously shown that BMP

[1]Institute of Biochemistry and Molecular Biology, Ulm University, Albert-Einstein-Allee 11, 89081 Ulm, Germany. [2]Department of Obstetrics and Gynecology, Ulm University, Prittwitzstraße 43, 89075 Ulm, Germany. [3]Department of Medical and Surgical Sciences, University of Bologna, via Massarenti 9, 40138 Bologna, Italy. [4]IRCCS Azienda Ospedaliero-Universitaria di Bologna, via Massarenti 9, 40138 Bologna, Italy. [5]Department of Dermatology and Allergic Diseases, Ulm University, Albert-Einstein-Allee 11, 89081 Ulm, Germany. [6]Institute of Molecular Medicine, Ulm University, Meyerhofstrasse N27, 89081 Ulm, Germany. [7]Department of Physiology, Anatomy and Genetics, University of Oxford, South Parks Road, Oxford OX1 3PT, United Kingdom. [8]Institute of Biomedical Ethics and History of Medicine, University of Zurich, Winterthurerstrasse 30, 8006 Zurich, Switzerland. [9]Institute of Experimental Cardiology, Heidelberg University, Im Neuenheimer Feld 669, 69120 Heidelberg, Germany. [10]Institute of Physiology I, Medical Faculty, University of Bonn, Venusberg-Campus 1, 53127 Bonn, Germany. [11]These authors contributed equally: Mohankrishna Dalvoy Vasudevarao, Denise Posadas Pena. ✉e-mail: gilbert.weidinger@uni-ulm.de

signaling is required for CM dedifferentiation and proliferation during heart regeneration, but not for CM proliferation during physiological heart growth[11]. This suggests that BMP signaling primarily acts on regeneration-specific cellular processes. Here we show that BMP signaling promotes CM regeneration by alleviation of replication stress.

Replication stress and other forms of DNA damage are considered leading causes of declining tissue renewal and repair in aged mammals[12–15]. Cells experience replication stress when DNA replication forks slow down or stall, due to unresolved DNA lesions like DNA strand breaks, collisions with transcriptional machinery or insufficient nucleotide supplies[16,17]. Replication stress can arise from the high cell cycling demands imposed by oncogenes or when aged stem cells are challenged to repair a tissue[17,18]. While cells can use several mechanisms to bypass or tolerate hindrances to replication, unresolved replication stress results in reduced tissue turnover and repair, functional decline of stem cells, cellular senescence, and apoptosis[16–20].

Here, we find that cycling CMs experience replication stress during zebrafish heart regeneration. We identify BMP signaling as a key factor that allows CMs to alleviate the stress, a function that is conserved in mammalian cells. Our findings add to the emerging view that highly regenerative organisms are not immune from challenges that impair regeneration in aged mammals. Rather, our work suggests that the ability to efficiently overcome replication stress is a key reason for the elevated capacity of adult zebrafish to regenerate the heart.

## Results

### Cardiomyocytes become γH2a.x positive during zebrafish heart regeneration

To identify mechanisms of heart regeneration, we analyzed the transcriptome of zebrafish hearts, and noticed that gene signatures associated with DNA repair are enriched at 7 days post injury (dpi), which represents the time-point of peak CM proliferation[1,3–5] (Fig. 1A, Supplementary data files 1 and 2). qRT-PCR confirmed that genes involved in several DNA damage response pathways are upregulated at 7 dpi in cryoinjured ventricles (Fig. 1B). CMs located at the wound border are key to heart regeneration, since they dedifferentiate and proliferate to replace the damaged myocardium[2]. Using single-cell sequencing data, we found that some of the upregulated DNA-repair associated genes were more strongly expressed in wound border-zone CMs than in remote CMs at 7 dpi (Fig. 1C). This prompted us to test whether elevated DNA damage can be detected in wound border CMs, using an antibody against a phosphorylated form of the histone variant H2a.x (γH2a.x), a well-established marker of DNA damage. Western blotting detected elevated levels of γH2a.x in ventricular tissue (from which the wound had been removed to enrich for CMs) at 7 dpi (Fig. 1D). Immunofluorescence showed that a sizable fraction of CMs at the wound border (within 150 μm of the wound) was positive for γH2a.x at 7 dpi (Fig. 1E). The temporal profile of γH2a.x-positivity closely resembled the profile for the fraction of CMs that are in the cell cycle, with numbers peaking at 7 dpi and reverting to baseline by 30 dpi (Fig. 1F). Furthermore, the number of γH2a.x-positive CMs decreased with distance from the wound (Fig. 1G) in a similar manner as reported for the fraction of cycling CMs[21]. γH2a.x-positive CMs were also observed in regenerating hearts that had been subjected to resection of the ventricular apex (Supplementary Fig. 1A). Together, these data suggest that CMs in regenerating hearts experience DNA damage that is independent of the type of injury and unlikely to be caused directly by the insult.

In mice, the general rise in oxygen tension after birth has been found to cause DNA damage that limits CM proliferation[22]. Thus, we wondered whether oxidative stress plays a role in inducing CM DNA damage in regenerating zebrafish hearts. We assessed oxidative DNA damage using ELISA for 8-hydroxy-2'-deoxyguanosine (8-OHdG). Peroxide treatment of heart explants induced DNA oxidation, but cryoinjury alone did not increase 8-OHdG levels compared to uninjured hearts (Fig. 1H). These data suggest that oxidative DNA damage is an unlikely cause of γH2a.x positivity in CMs.

### Cardiomyocytes experience replication stress during regeneration but not physiological heart growth

The temporal and spatial profiles of γH2a.x accumulation suggested that CMs that enter the cell cycle at the wound border experience DNA damage. To test this, we labeled cycling CMs using repeated injection of EdU and stained for γH2a.x and EdU at 7 dpi (Fig. 2A). We found that ~55% of the EdU+ CMs were also positive for γH2a.x. Conversely, 81% of the γH2a.x+ CMs were EdU-positive, showing that γH2a.x+ was largely confined to CMs that had recently been cycling or were still in a cycling state. These data suggest that CMs become γH2a.x positive because they experience replication stress when they enter the cell cycle during regeneration. To test whether cycling CMs accumulate γH2a.x+ also during physiological heart growth, we labeled CMs in uninjured hearts of juvenile fish (30 days post fertilization [dpf], 16 mm standardized standard length, [SSL]) with an 8 h EdU pulse and stained for γH2a.x. We found that only 0.62% of CMs were γH2a.x +, while treatment with hydroxyurea (HU), which depletes cells of nucleotides and thus causes replication stress, did induce γH2a.x+ accumulation (Supplementary Fig. 2A). Double staining for EdU and γH2a.x revealed that γH2a.x accumulation was rare among the cycling CMs (2.3%), but could be strongly increased by HU treatment (Fig. 2B). Similarly, in adult cryoinjured hearts, HU treatment increased the fraction of γH2a.x+ CMs (Supplementary Fig. 2B), while it reduced CM cell cycling (Supplementary Fig. 2C) and the fraction of γH2a.x+ CMs that were also EdU+ (Supplementary Fig. 2D). Together, these data show that γH2a.x does not generally stain cycling zebrafish cardiomyocytes, but serves as a reliable readout for replication stress. The subnuclear localization of γH2a.x is also indicative of the type of DNA damage[23]; ionizing irradiation, which causes double-strand breaks primarily, resulted in the accumulation of γH2a.x in distinct puncta in CMs of adult regenerating hearts, while HU caused pan-nuclear γH2a.x staining (Supplementary Fig. 2E). Importantly, 84% of the γH2a.x+ CMs of unperturbed regenerating hearts displayed pan-nuclear γH2a.x distribution (Fig. 2C). Accumulation of phosphorylated Rpa32, which forms part of the RPA-complex of single-stranded DNA binding proteins, is considered a specific readout for replication stress[24]. Western blotting showed that p-Rpa32 (S33) levels strongly increased by 7 dpi in myocardial ventricular tissue (Fig. 2D). We conclude that wound border CMs experience replication stress, which can be detected by γH2a.x accumulation.

If heart regeneration induces replication stress, the fraction of γH2a.x+ CMs should be sensitive to interventions that increase or decrease the demand for regenerative CM cell cycling. A single injection of the Vitamin D analog calcidiol, which enhances heart regeneration[25], increased the fraction of mitotic phospho-Histone H3+ (pH3) wound border CMs at 7 dpi and resulted in a strong trend towards an increase in the fraction of γH2a.x+ CMs as well (Supplementary Fig. 3A, B). mTOR signaling represents one of the pathways required for CM cycling during zebrafish heart regeneration[26]. We found that treatment with the mTOR inhibitor rapamycin reduced both the fraction of cycling and the fraction of γH2a.x+ CMs at 7 dpi (Supplementary Fig. 3C). Similarly, starvation yielded comparable results (Supplementary Fig. 3D). To further support the correlation between CM cycling and γH2a.x accumulation in regenerating hearts, we directly inhibited cell cycle progression using the CDK4/6 inhibitor PD-0332991. One day treatment was sufficient to strongly decrease the number of mitotic pH3 CMs and the number of γH2a.x+ CMs at 7 dpi (Fig. 2E, F). In juvenile fish, the rate of heart growth and thus of physiological CM cycling can be modulated by keeping fish at different densities[27]. Intriguingly, shifting juvenile fish (42 dpf, 11–16 mm SSL) from restricted to stimulated growth conditions did increase the

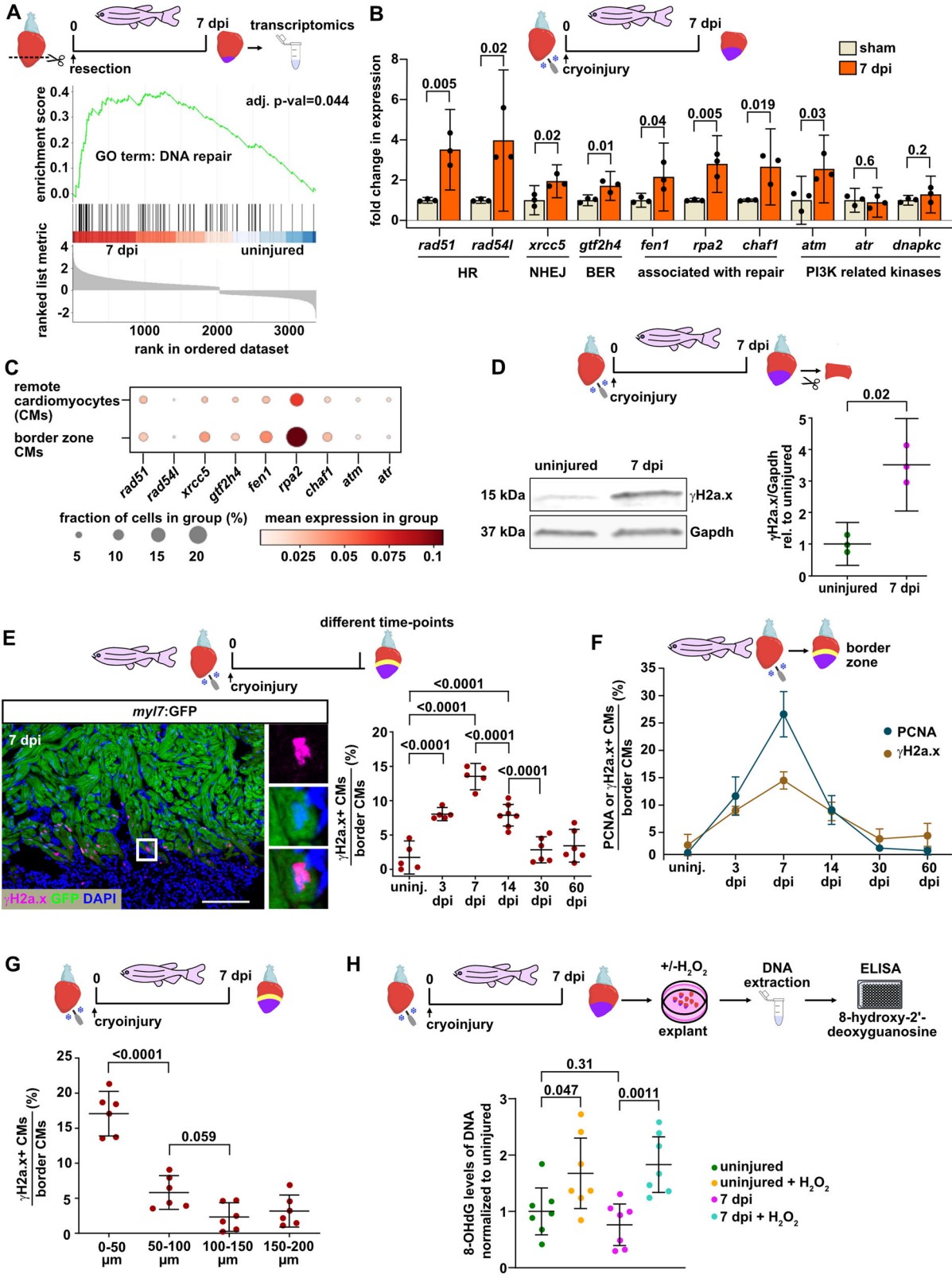

fraction of cycling CMs, but did not induce γH2a.x+ accumulation (Fig. 2G). However, γH2a.x+ accumulated in CMs of juvenile fish when their hearts were injured (Fig. 2H). Together, these findings support the conclusion that cardiomyocyte replication stress is specifically induced by the demands of regeneration, but not by physiological growth.

In addition to cell cycle entry, wound border CMs undergo dedifferentiation, that is they downregulate characteristics of the differentiated state and re-express embryonic markers[2]. We thus wondered whether replication stress impinges on CM dedifferentiation. Induction of exogenous replication stress via HU treatment did not affect three readouts of CM dedifferentiation, namely the upregulation of a

**Fig. 1 | Signatures of DNA damage are induced in regenerating zebrafish cardiomyocytes (CMs). A** Gene Set Enrichment Plot showing enrichment of transcripts associated with the GO term "DNA repair" in zebrafish hearts at 7 days post ventricular resection injury (dpi). Fold changes were calculated with DESeq2 using the Wald test. **B** qRT-PCR confirms upregulation of DNA damage response genes in ventricles at 7 days post cryoinjury (dpi), relative to sham-treated hearts. $n_E$ (biological replicates) = 3, $n_A$ (animals) = 4 per replicate. Numbers indicate *p*-value. HR, homologous recombination; NHEJ, non-homologous end-joining; BER, base excision repair. **C** Dot plot visualization of single cell RNASeq expression data of border zone and remote CM clusters at 7 dpi for the genes upregulated in (**B**). Source data is available in: GSE130940. **D** Western blotting of wound border zone tissue shows increased levels of γH2a.x at 7 dpi relative to uninjured hearts. $n_E$ = 3, $n_A$ = 10 per replicate. **E** Immunofluorescence on cryosections of *myl7*:GFP transgenic hearts reveals γH2a.x accumulation in GFP+ CMs located within 150 μm from the wound border at 7 dpi. White box, magnified region. $n_E$ = 1, $n_A$ = 5 for uninjured, 3 and 7 dpi

groups; 7 for 14 dpi: 6 for 30 and 60 dpi groups, $n_C$ (analyzed CMs) = 38200 across all time-points. Scale bar, 100 μm. **F** The fraction of γH2a.x+ wound border CMs correlates with the fraction of cycling CMs. Fraction of γH2a.x+ CMs presented in (**E**) is overlaid with data of PCNA+ cycling CMs as published in ref. 7. $n_E$ = 1, $n_A$ = 5 for uninjured, 3 and 7 dpi groups; 7 for 14 dpi: 6 for 30 and 60 dpi groups, $n_C$ (analyzed CMs) = 38200 across all time-points. **G** γH2a.x+ CMs are found close to the wound border at 7 dpi. $n_E$ = 1, $n_A$ = 6, $n_C$ = 16400 total. **H** Oxidative DNA damage is induced by hydrogen peroxide treatment of explanted hearts, but does not differ between uninjured and regenerating hearts as determined by 8-OHdG ELISA at 7 dpi. $n_E$ = 2, $n_A$ = 7 per group. **B–H** Two tailed Student´s *t*-test; (**E**, **F**) Ordinary one-way ANOVA with Bonferroni correction; (**G**) Ordinary one-way ANOVA with Dunnet´s multiple comparison test; (**A–H**) Data are presented as mean values. Error bars indicate confidence interval (CI) 95%. Source data are provided in the Source Data file. Some elements were created in BioRender (Agreement number: AH27UPOG7H; Posadas, D. (2025) https://BioRender.com/w88l548).

transgene reporting the activity of regulatory regions of the CM progenitor marker *gata4*, the upregulation of an embryonic myosin (embMHC) or the disassembly of sarcomeres (Supplementary Fig. 4A, B). This indicates that CM dedifferentiation occurs independently and likely upstream of CM cell cycling, which is reduced by HU.

## CM replication stress is likely not due to the accumulation of DNA lesions or collisions with transcription
Replication stress can be caused by a variety of molecular reasons. One option is that regenerating CMs are particularly sensitive towards blockade of replication forks by DNA lesions that already exist in the DNA prior to S-phase. If so, the fraction of γH2a.x+ CMs might increase in injured hearts with age of the fish, since lesions should accumulate with age. Yet, we did not observe such an increase in the fraction of γH2a.x+ CMs in regenerating hearts between juvenile stages (42 days), early-mid adulthood (6 months) and aged conditions (2 years old) (Fig. 2H). This indicates that CM replication stress is unlikely to be caused by blockade of replication forks by pre-existing DNA lesions. Collision of replication forks with RNA transcription represents another cause of replication stress[17]. Yet, immunostaining against the active, elongating phosphorylated form of RNA Polymerase II showed that overall levels of transcription appear to be lower in wound border CMs compared to other regions of the ventricle, making it unlikely that proliferating CMs experience increased conflicts between transcription and replication (Supplementary Fig. 5A).

## DNA damage signaling is required for heart regeneration
We next asked whether CMs that experience replication stress become senescent or undergo apoptosis. At 7 dpi, the senescence marker beta-galactosidase was upregulated in the wound area in what appeared by position to be epicardial cells, but not in CMs (Supplementary Fig. 5B), in agreement with previous reports in regenerating zebrafish and neonatal mouse hearts[28,29]. Interestingly, induction of exogenous replication stress by HU treatment for 7 days was not sufficient to induce CM senescence by 7 dpi (Supplementary Fig. 5C). We conclude that CMs that experience replication stress do not becoming senescent. In unperturbed regenerating hearts, only 0.27% of the wound border CMs were apoptotic as detected by Caspase3 accumulation, while ionizing radiation induced CM apoptosis (Fig. 3A). These results suggest that zebrafish CMs experiencing replication stress are eventually able to overcome this stress and continue to proliferate, despite having retained the capacity to induce pro-apoptotic signaling in response to DNA damage. If so, DNA damage pathways that sense replication stress might be required for heart regeneration. We first tested whether inhibitors of the ATM and ATR kinases, which mediate cellular responses to DNA-damage and replication stress, respectively, are functional in zebrafish. We found that the ATM inhibitor KU55933 and the ATR inhibitor VE821 were non-toxic, since they did not affect embryonic development (Supplementary Fig. 6A, B). Yet, embryos

treated with the ATR inhibitor displayed necrosis when exposed to doses of HU that on their own had little effect (Supplementary Fig. 6A), while the ATM inhibitor exacerbated the effect of low doses of ionizing radiation (Supplementary Fig. 6B). These data show that these inhibitors block the ability of cells to activate repair pathways in response to replication stress and other forms of DNA damage in zebrafish. Replication stress is generally thought to be mainly sensed by ATR activation[30]. Intriguingly, we found that treatment of fish with the ATR inhibitor VE821 for 24 h was sufficient to reduce CM mitosis in regenerating hearts at 7 dpi (Fig. 3B). It is known that the ATM and ATR kinase pathways can crosstalk to promote replication stress responses[31]. Indeed, combined treatment with the ATM and ATR inhibitors suppressed CM mitosis more strongly than ATRi alone (Fig. 3C). Treatment with both inhibitors in combination also impaired morphological heart regeneration, as quantified by shrinkage of the wound between 3 and 21 dpi (Fig. 3D). Together, these data show that activation of DNA damage repair pathways, mainly mediated by ATR, but with contributions from ATM, is essential for regenerative CM proliferation and heart regeneration.

## BMP signaling alleviates CM replication stress
Our data suggest that alleviation of replication stress is essential for efficient regeneration of CMs. We thus wondered whether specific components of the pro-regenerative signaling environment of the zebrafish heart promote regeneration by impinging on CM replication stress. Likely candidates are pathways that regulate regenerative, but not physiological CM proliferation. We have previously shown that BMP signaling is activated in wound border CMs during zebrafish heart regeneration, but not under conditions of physiological growth, and that it is essential for regenerative CM proliferation[11]. Therefore, we asked whether BMP signaling modulates CM replication stress. Using a heat-shock-inducible transgene (*hsp70l*:nog3[fr14Tg]), we overexpressed the secreted inhibitor *noggin3* (*nog3*), which interferes with binding of BMP ligands to their receptors and blocks both Smad-dependent and Smad-independent non-canonical BMP signaling (Fig. 4A). Compared to heat-shocked wild-type fish, *nog3*-expressing fish displayed a significantly higher fraction of γH2a.x+ CMs at 7 dpi (Fig. 4B), whereas overexpression of the BMP ligand *bmp2b* (using *hsp70l*:bmp2b[fr13Tg] fish) very efficiently reduced the number of γH2a.x+ CMs (Fig. 4B). Importantly, a single pulse of induced *bmp2b* overexpression was sufficient to strongly reduce the number of γH2a.x+ CMs within 6 h (Fig. 4C), suggesting that BMP signaling can directly act on cycling CMs that experience replication stress.

Heterodimers of Bmp2 or Bmp4 with Bmp7 seem to be the most potent ligands in many contexts, suggesting that Bmp7 is the least redundant of these molecules[32,33]. Indeed, overexpression of *bmp7b* or *bmp4* was also sufficient to reduce the fraction of γH2a.x+ CMs (Fig. 4D and Supplementary Fig. 7A). The fraction of γH2a.x+ CMs at 7 dpi was increased in fish homozygous mutant for *bmp7a*, one of the two *bmp7*

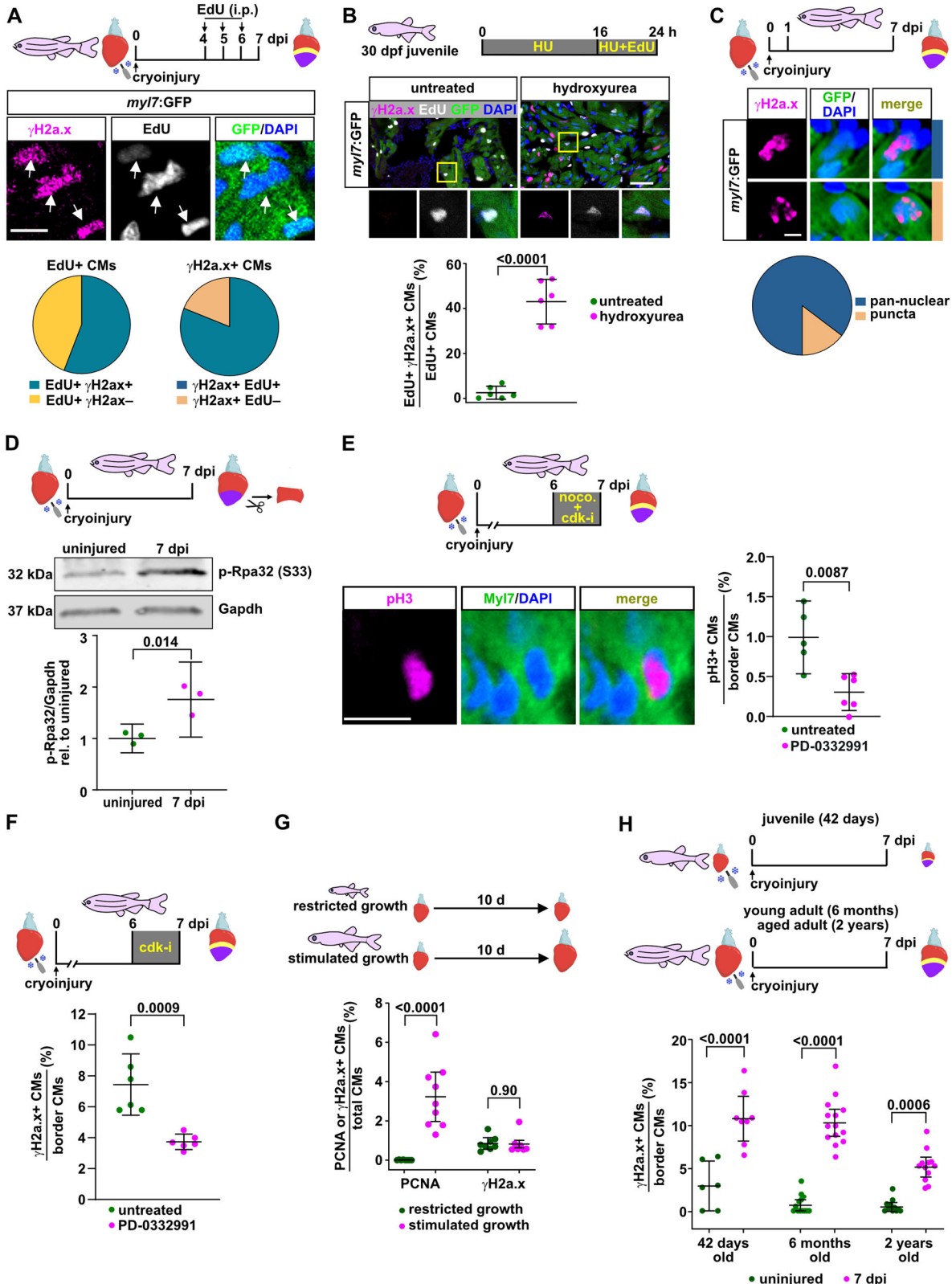

paralogs in zebrafish (Fig. 4E), while p-Smad1/5/9 levels were decreased (Supplementary Figs. 7B) in the absence of morphological differences between mutant and wild-type hearts (Supplementary Fig. 7C). We conclude that endogenous BMP signaling is required for alleviation of CM replication stress during regeneration, and that Bmp7a acts as non-redundant ligand in this process. We also asked whether BMP signaling can alleviate additional, exogenously induced

replication stress. Intriguingly, in HU-treated fish, *nog3* overexpression further increased the fraction of γH2a.x+ CMs, while *bmp2b* overexpression strongly reduced it (Fig. 4F).

## BMPs act through Smads to curb replication stress

BMP ligands can activate Smad-dependent and -independent (non-canonical) signaling pathways (Fig. 5A)[34]. Since non-canonical

**Fig. 2 | CM replication stress is a specific feature of regeneration and not of physiological growth or aging. A** Immunofluorescence for γH2a.x and GFP combined with EdU detection in *myl7*:GFP transgenic hearts at 7 dpi. Cycling CMs were labeled by daily EdU injections from 4 to 6 dpi. $n_E = 1$, $n_A = 6$, $n_C = 7170$. Scale bar, 10 μm. Arrows point to EdU+ γH2a.x+ CM nuclei. **B** In untreated uninjured juvenile fish at 30 days post fertilization (30 dpf), cycling CMs (labeled by immersion of fish in EdU for 8 h) are not γH2a.x + while hydroxyurea (HU) treatment induces γH2a.x+ accumulation. Yellow box, magnified region. $n_E = 1$, $n_A = 6$ per treatment, $n_C = 4560$ untreated, 3660 HU. Scale bar, 50 μm. **C** Most of γH2a.x+ CMs at the wound border at 7 dpi present pan-nuclear staining (upper image) versus distinct puncta (lower image). $n_E = 1$, $n_A = 6$, $n_C = 6450$. Scale bar, 10 μm. **D** Western blotting shows increased levels of p-Rpa32 (S33) in the wound border at 7 dpi. $n_E = 3$, $n_A = 10$ per replicate. **E** Treatment of cryoinjured fish with the cdk4/6 inhibitor PD-0332991 for 1 d reduces the fraction of mitotic pH3+ wound border CMs at 7 dpi. Nocodazole treatment was used to block cytokinesis, increasing detectable

pH3+ cells. $n_E = 1$, $n_A = 5$ untreated, 6 treated, $n_C = 10114$ untreated, 11826 treated. Scale bar, 10 μm. **F** PD-0332991 treatment decreases the fraction of γH2a.x+ wound border CMs. $n_E = 1$, $n_A = 6$ per treatment, $n_C = 2352$ untreated, 2688 treated. **G** Stimulated growth condition increases the fraction of PCNA+ CMs without an associated increase in γH2a.x+ CMs in uninjured juvenile fish at 30 dpf (16 mm SSL). $n_E = 2$ for PCNA and 1 for γH2a.x, $n_A = 8$ restricted growth (RG) PCNA, 8 stimulated growth (SG) PCNA, 8 RG γH2a.x, 7 SG γH2a.x. $n_C = 83980$ RG, 95600 SG. **H** The fraction of wound border γH2a.x+ CMs at 7 dpi does not increase with fish age. $n_E = 1$ (42 days), 2 for 6 months and 2 for 2 years, $n_A = 6$ (42 days uninjured), 8 (42 days, 7 dpi), 13 (6 months uninjured), 14 (6 months, 7 dpi), 11 (2 years, uninjured), 12 (2 years, 7 dpi), $n_C = 31700$ (total for all conditions). **A**–**H** Data are presented as mean values. Error bars, CI 95%. Two tailed Student´s *t*-test. Source data are provided in the Source Data file. Some elements were created in BioRender (Agreement number: AH27UPOG7H; Posadas, (2025) https://BioRender.com/w88l548).

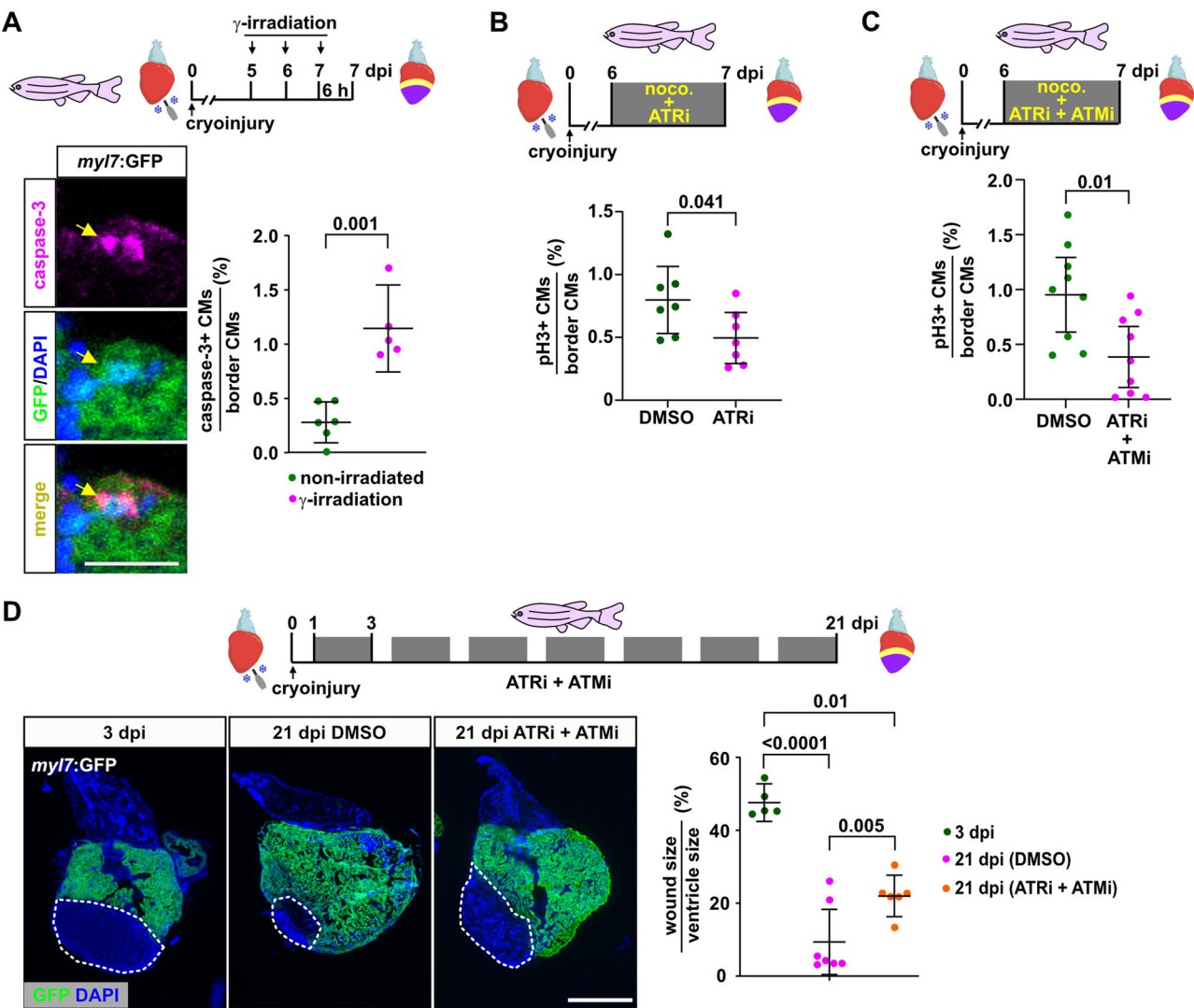

**Fig. 3 | DNA damage repair is essential for zebrafish CM regeneration. A** Very few caspase-3+ apoptotic CMs can be detected at the wound border at 7 dpi, while γ-irradiation induces CM apoptosis. Arrow points to perinuclear caspase-3+ areas in a single apoptotic CM. $n_E = 1$, $n_A = 6$ non-irradiated, 5 γ-irradiation, $n_C = 3930$ non-irradiated, 5950 irradiated. **B** Treatment of cryoinjured fish with the ATR inhibitor VE821 for 24 h reduces the fraction of pH3+ wound border CMs in *myl7*:H2b-GFP transgenic hearts at 7 dpi. $n_E = 1$, $n_A = 7$ per condition, $n_C = 18400$ DMSO, 21100 ATRi. **C** Combined treatment with the ATR and ATM inhibitors VE821 and KU55933 for 24 h reduces wound border CM mitosis. $n_E = 2$, $n_A = 9$ per treatment, $n_C = 16720$

DMSO, 15610 ATRi + ATMi. **D** Combined intermittent treatment (2 days on, 1 day off) of cryoinjured fish with VE821 and KU55933 from 1 dpi to 21 dpi results in a reduction in wound resorption compared to DMSO treated fish. Wound size (dotted lines) is determined by area lacking GFP expression in *myl7*:GFP transgenics. $n_E = 1$, $n_A = 5$ for 3 dpi, 8 for 21 dpi DMSO and 7 for 21 dpi ATRi + ATMi. Scale bar, 250 μm. **A**–**D** Data are presented as mean values. Error bars, CI 95%. Two tailed Student´s *t*-test. Source data are provided in the Source Data file. Some elements were created in Biorender (Agreement number: AH27UPOG7H; Posadas, (2025) https://BioRender.com/w88l548).

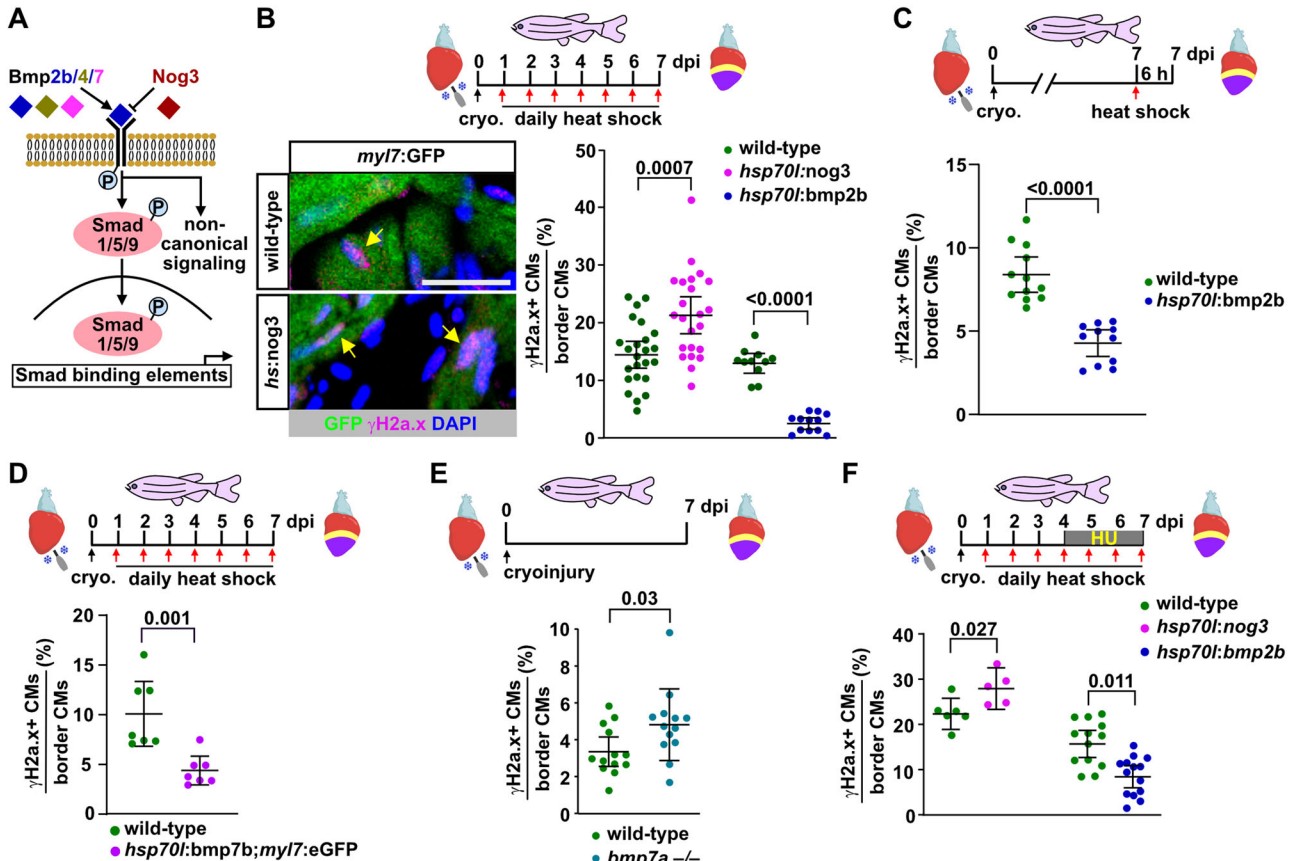

**Fig. 4 | BMP signaling alleviates CM replication stress during zebrafish heart regeneration. A** Cartoon depicting BMP signaling. Heat-shock-induced overexpression of the BMP ligands 2, 4 or 7 from transgenes activates Smad-mediated and "noncanonical" signaling pathways (gain-of-function, GOF), while overexpression of the secreted inhibitor Noggin3 (Nog3) inhibits them (loss-of-function, LOF). **B** BMP-LOF caused by overexpression of *noggin3* via daily heat-shock from 1 to 7 dpi increases the fraction of γH2a.x+ CMs at 7 dpi relative to heat-shocked wild-type siblings, while BMP-GOF via *bmp2b* overexpression strongly reduces it. Double transgenics with *myl7*:GFP were used to identify CMs, "wild-type" refers to absence of the BMP modifying transgene. $n_E = 3$, $n_A = 24$ wild-type (wt) siblings to *hsp70l*:nog3, 23 *hsp70l*:nog3, 11 wt siblings to *hsp70l*:bmp2b, 12 *hsp70l*:bmp2b, $n_C = 41460$ total across all groups. Scale bar, 50 µm. **C** BMP-GOF is sufficient to reduce the fraction of γH2a.x+ CMs within 6 h after a single heat-shock of *hsp70l*:bmp2b transgenics at 7 dpi. $n_E = 2$, $n_A = 12$ wt, 11 *hsp70l*:bmp2b,

$n_C = 4872$ wt, 6470 *hsp70l*:bmp2b. **D** *bmp7b* overexpression reduces the fraction of γH2a.x+ CMs. $n_E = 1$, $n_A = 7$ per group, $n_C = 4900$ wt, 3500 *hsp70l*:bmp7b;*myl7*:GFP. **E** *bmp7a* −/− mutants display an increased fraction of γH2a.x+ CMs at 7 dpi compared to wild-type siblings. $n_E = 2$, $n_A = 13$ per group, $n_C = 7800$ wt, 7600 *bmp7a* −/−. **F** BMP-LOF using *hsp70l*:nog3 enhances exogenous, HU-induced replication stress in border zone CMs, while BMP-GOF using *hsp70l*:bmp2b transgenics rescues CMs from HU-induced γH2a.x accumulation. $n_E = 1$ for wt vs *hsp70l*:nog3 and 2 for wt vs *hsp70l*:bmp2b, $n_A = 6$ wt siblings to *hsp70l*:nog3, 5 *hsp70l*:nog3, 13 wt siblings to *hsp70l*:bmp2b, 14 *hsp70l*:bmp2b, $n_C = 47860$ total CMs across all groups. **B**–**F** Data are presented as mean values. Error bars, CI 95%. Two tailed Student´s *t*-test. Source data are provided in the Source Data file. Some elements were created in Biorender (Agreement number: AH27UPOG7H; Posadas, (2025) https://BioRender.com/w88l548).

BMP-activated pathways have been described to affect the response of mammalian CMs to heart injury[35], we asked whether BMPs act through Smads in the alleviation of replication stress in the zebrafish heart. To test this, we created a transgenic line in which the inhibitory Smad6b, together with nuclear Tomato (nT), is expressed after heat-shock (*hsp70l*:nT-p2a-smad6b[ulm16Tg]). Expression of the transgene in adult CMs was mosaic after a single heat-shock, which allowed us to compare nT+ to nT− CMs. Smad signaling as read-out by nuclear accumulation of phosphorylated Smad1/5/9 was suppressed in nT+ CMs within 6 h post heat-shock at 7 dpi (Fig. 5B). Intriguingly, nT+ CMs were also 3 times more frequently positive for γH2a.x than wild-type or nT− CMs (Fig. 5C). We conclude that BMP/Smad signaling is required cell-autonomously in CMs for alleviation of replication stress. To test whether overexpressed BMP ligands act solely through the Smad pathway in this context, we analyzed fish double transgenic for *hsp70l*:bmp2b and *hsp70l*:nT-p2a-smad6b. As expected, the fraction of p-Smad1/5/9+ CMs was increased upon *bmp2b* overexpression, and decreased by *smad6b*, but in double transgenic

fish the fraction of p-Smad1/5/9+ CMs was comparable to that seen in wild-types (Supplementary Fig. 8A). This allowed us to ask whether co-expression of *smad6b* can reverse the effect of *bmp2b* overexpression on γH2a.x. While fish that only expressed *bmp2b* showed a strong reduction in the number of γH2a.x+ CMs, this effect was completely abrogated by the co-expression of *smad6b* (Fig. 5D). Together, these data indicate that BMP signaling acts directly in CMs and through Smads to alleviate replication stress. To probe the long-term consequences of inhibition of BMP/Smad signaling on morphological heart regeneration and scarring, we subjected cryoinjured *hsp70l*:nT-p2a-smad6b transgenics and wild-type siblings to daily heat-shocks for 21 days and assessed wound size and collagen deposition using acid fuchsin orange G (AFOG) staining. While wound sizes did not differ at 3 dpi, *hsp70l*:nT-p2a-smad6b cryoinjured hearts displayed larger wounds at 21 dpi (Supplementary Fig. 8B). Importantly, these wounds contained considerably more collagen (Fig. 5E). We conclude that BMP/Smad signaling is required for morphological heart regeneration and scar resolution.

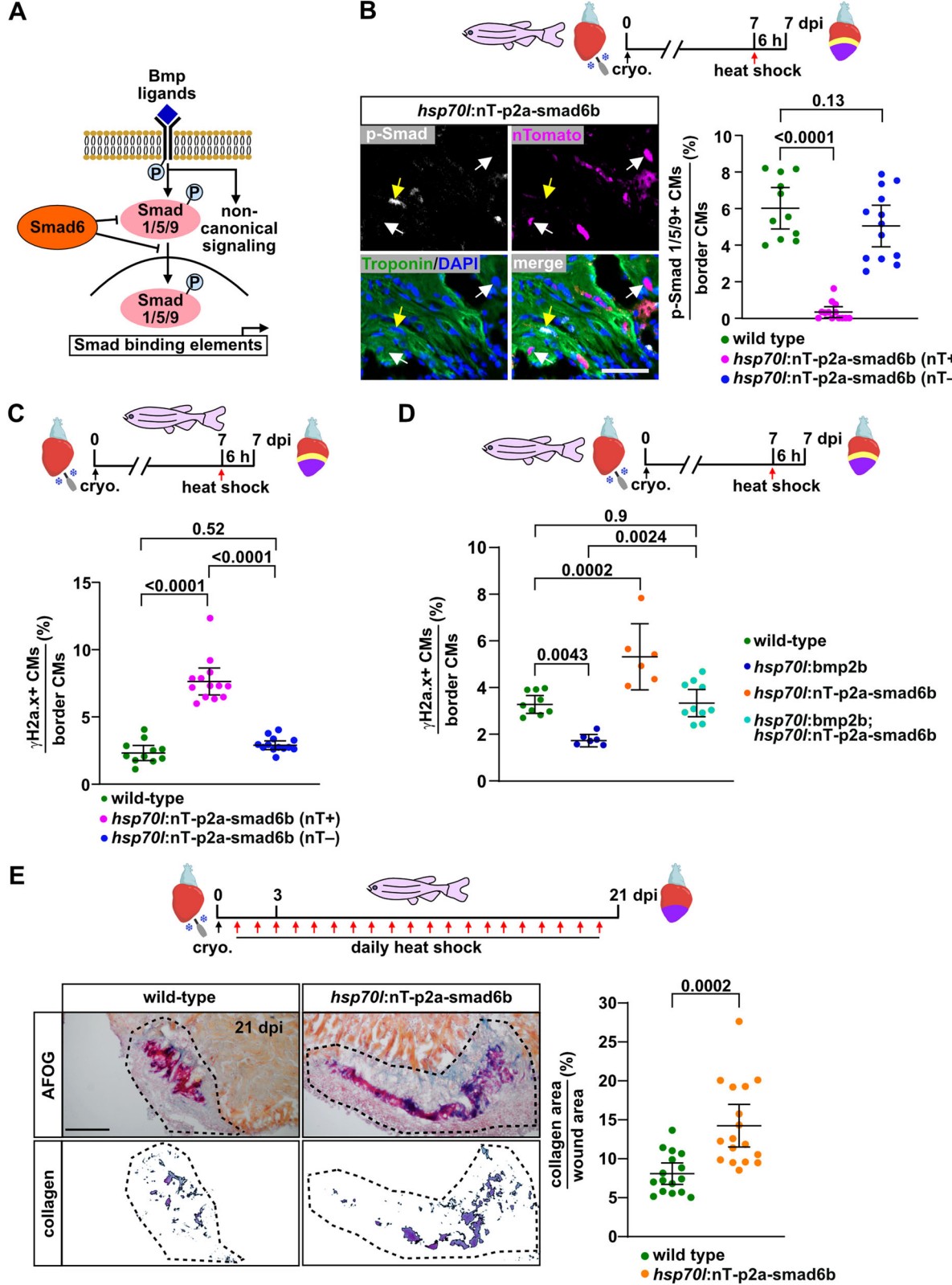

## BMP signaling promotes stress-free replication and progression into mitosis

Of note, BMP signaling seems to behave differently than other interventions that alter CM cell cycling and proliferation. As shown above, the fraction of cycling CMs is closely positively correlated with those that are γH2a.x+ across various experimental interventions, including inhibition of mTOR and activation of Vitamin D

signaling (see Fig. 2). In contrast, loss of *bmp7a* decreases CM cycling[36], but increases the fraction of γH2a.x+ CMs (see Fig. 4). This prompted us to further explore how BMP signaling regulates regenerative CM cycling and proliferation.

A single pulse of *smad6b* expression reduced the fraction of pH3+ mitotic CMs in cryoinjured *hsp70l*:nT-p2a-smad6b fish within 6 h (Fig. 6A). To test if cells experience cell cycle delay in the absence of

**Fig. 5 | BMP-mediated alleviation of replication stress requires Smad signaling.**
**A** Cartoon depicting specific inhibition of the canonical BMP-Smad pathway via overexpression of the inhibitory *smad6b*. **B** Immunofluorescence for nuclear Tomato (nT, white arrows) reveals mosaic expression of the *hsp70l*:nT-p2a-smad6b transgene 6 h after a single heatshock at 7 dpi. p-Smad1/5/9 immunostaining reveals inhibition of BMP-Smad signaling in nT+ CMs (identified by Troponin staining), while nT− CMs in transgenic hearts are equally likely to be p-Smad1/5/9+ than CMs in heat-shocked wild-type fish. $n_E = 2$, $n_A = 11$ wt, 13 *hsp70l*:nT-p2a-smad6b, $n_C = 6400$ wt, 4540 nT + , 2950 nT−. Scale bar, 100 µm. **C** BMP-Smad LOF using *hsp70l*:nT-p2a-smad6b increases γH2a.x accumulation in nT+ CMs compared with wild-type or nT− CMs after a single heat-shock 6 h before harvest at 7 dpi. $n_E = 2$, $n_A = 11$ wt, 13 *hsp70l*:nT-p2a-smad6b, $n_C = 5800$ wt, 2600 nT + , 2780 nT−. **D** Smad6 co-expression blocks the ability of BMP2b GOF to alleviate CM replication stress, since the fraction of γH2a.x+ CMs in *hsp70l*:nT-p2a-smad6b; *hsp70l*:bmp2b double transgenics is not different from that in heat-shocked wild-type fish 6 h after

a single heat-shock at 7 dpi. $n_E = 2$, $n_A = 9$ wt, 6 *hsp70l*:bmp2b, 6 *hsp70l*:nT-p2a-smad6b, 10 *hsp70l*:nT-p2a-smad6b and *hsp70l*:bmp2b, $n_C = 13250$ total across all groups. The observed relative difference between wild-type and double transgenics is 2%, the calculated smallest significant difference 22%, which is smaller than the difference between *hsp70l*:bmp2b and double transgenics of 47%. Thus, this experiment had enough power to reveal biologically relevant effects. **E** AFOG staining reveals increased wound collagen content in *hsp70l*:nT-p2a-smad6b transgenics at 21 dpi following daily heat-shock. Myocardium, brown; collagen, blue; fibrin, red. Lower images show masks used to quantify the blue collagen+ areas. $n_E = 2$, $n_A = 16$ wt, 17 *hsp70l*:nT-p2a-smad6b. Scale bar, 50 µm. **B–D** Ordinary one-way ANOVA with Bonferroni correction; Two tailed Student´s *t*-test. **B–E** Data are presented as mean values. Error bars, CI 95%. Source data are provided in the Source Data file. Some elements were created in Biorender (Agreement number: AH27UPOG7H; Posadas). (2025) https://BioRender.com/w88l548).

BMP signaling, we labeled cycling CMs of injured wild-type and *hsp70l*:nog3 transgenic hearts for 3 days via repeated EdU injections, and stained for EdU and PCNA at 7 dpi (Fig. 6B). Our previous quantification and modeling of CM number increase during regeneration has suggested that individual CMs do not undergo multiple rounds of cell division[7]; thus, a minority of CMs that are cycling at 4, 5 or 6 dpi are expected to still be cycling at 7 dpi, rather many will have withdrawn from the cell cycle by that time. Building on this concept, we define EdU+ PCNA+ CMs at 7 dpi as those with ongoing cell cycle, which are possibly delayed in exiting the cycle, while CMs that are only EdU+ as those that have completed cycling (Fig. 6B). Indeed, in heat-shocked wild-type fish only ~35% of the EdU+ CMs were PCNA+ at 7 dpi (Fig. 6B). Interestingly, inhibition of BMP signaling via overexpression of *nog3* increased the fraction of EdU+ PCNA+ CMs (Fig. 6B). *smad6b* overexpression increased the fraction of PCNA+ CMs within 6 h after a single pulse of expression (Supplementary Fig. 8C) but reduced CM mitosis (see Fig. 6A), strongly suggesting that *smad6b* induces CM cell cycle delay, which leads to an increase in the population of CMs with ongoing cell cycle. To test whether CMs were indeed stuck in S- or G2-phase because they could not overcome replication stress in the absence of BMP signaling, we again labeled cycling CMs by repeated EdU injection and then stained for EdU and γH2a.x (Fig. 6C). We reasoned that EdU+ γH2a.x+ CMs experience replication stress, while CMs that were only EdU+ enjoyed stress-free replication (Fig. 6C). We found that the fraction of EdU+ γH2a.x+ stressed CMs was increased upon overexpression of *nog3* (Fig. 6C). Interestingly, also *bmp7a* −/− fish displayed a significantly higher fraction of γH2a.x+ EdU+ stressed CMs than wild-type siblings (Fig. 6D). Overall, these data suggest that BMP signaling specifically promotes CM proliferation because it allows CMs to overcome replication stress and to proceed into mitosis.

Next, we wondered whether BMP gain-of-function has the opposite effect, namely whether it can promote CM cell cycle progression and stress-free replication. While *bmp2b* overexpression was not sufficient to decrease the fraction of PCNA+ EdU+ CMs in otherwise unperturbed conditions, it alleviated the increase in their numbers caused by HU treatment (Fig. 6E). Intriguingly, *bmp2b* overexpression was able to increase the fraction of EdU+ γH2a.x− CMs that experience "stress-free" replication both under unperturbed conditions and upon additional replication stress induced by HU (Fig. 6F).

## BMP signaling protects mouse cardiomyocytes and human cells from replication stress

We went on to test whether the ability of BMP signaling to alleviate replication stress is conserved in mammals. For a short period of time after birth, mice can regenerate the heart, and CMs isolated from pups at postnatal day 1 (P1) retain proliferative ability in culture[37]. We have previously shown that BMP7 promotes cell cycle progression and cell division in cultured mouse CMs[36]. While treatment with either recombinant BMP2 or BMP7 protein did not affect the baseline

number of γH2a.x+ CMs in the absence of HU, it abrogated the ability of HU to induce γH2a.x accumulation (Fig. 7A). We conclude that BMP2 and BMP7 protect neonatal mouse CMs from HU-induced replication stress. Several signaling pathways enhance mammalian CM proliferation, including inhibition of p38 kinase activity[38,39]. Yet, the p38 kinase inhibitor SB202190 did not interfere with the ability of HU to induce replication stress (Fig. 7B). We conclude that the ability to alleviate replication stress is a specific feature of BMP signaling. Next, we asked whether BMP signaling can protect human cells from replication stress. In the human U2OS cell line, combined treatment with recombinant BMP2 and BMP4 proteins was sufficient to completely abrogate the induction of γH2a.x in response to HU treatment (Fig. 7C). Likewise, BMP2 and BMP4 treatment suppressed the ability of HU to induce γH2a.x in neonatal human foreskin dermal fibroblasts (Fig. 7D), while it did not enhance replication as observed by EdU incorporation (Supplementary Fig. 9A). We also asked whether BMP signaling can protect hematopoietic stem cells from replication stress-induced functional decline, which has been shown to occur in aged mice[18]. Treatment with HU for 4 h significantly reduced the ability of primary human bone marrow CD34+ hematopoietic stem and progenitor cells (HSPCs) to form clonal colonies in a colony forming unit (CFU) assay[40]. Pre-treatment with BMP2 for just 2 h prior to HU treatment was sufficient to rescue CFU counts to levels of non-HU treated HSPCs (Fig. 7E). In summary, these data strongly suggest that alleviation of replication stress is a feature of BMP signaling that is conserved over a range of cell types and within vertebrate species, including human stem and progenitor cells.

## BMP signaling can speed the progression of replication forks and enhances fork re-start

To test whether BMP signaling acts directly on DNA replication, we turned to DNA fiber spreading assays in cultured human cord-blood HSPCs and the U2OS cancer cell line. In highly proliferative cells in culture, short incubation with nucleotide analogs (CldU followed by IdU), followed by imaging of individual DNA molecules and quantification of the length of labeled stretches of DNA ("tracks") can reveal several aspects of replication dynamics, including how fast replication forks progress[41]. Both cell types encounter replication challenges even under unperturbed culture conditions, HSPCs due to enforced S-phase entry and U2OS cancer cells due to oncogene overexpression[42–44]. Ongoing replication forks can be identified by dual-color tracks where CldU staining is followed by IdU staining (Fig. 8A). Treatment with BMP2 or BMP4 recombinant proteins was sufficient to increase the CldU and IdU track lengths of ongoing replication forks in HSPCs (Fig. 8A), while in U2OS cells only BMP4 had this effect (Fig. 8B). Replication fork stalling can be identified as tracks where the CldU+ and IdU+ stretches are of different lengths and can be quantified as the ratio between the longer and shorter stretches[41]. Except for a very small change caused by BMP2 in HSPCs (long/short track ratio

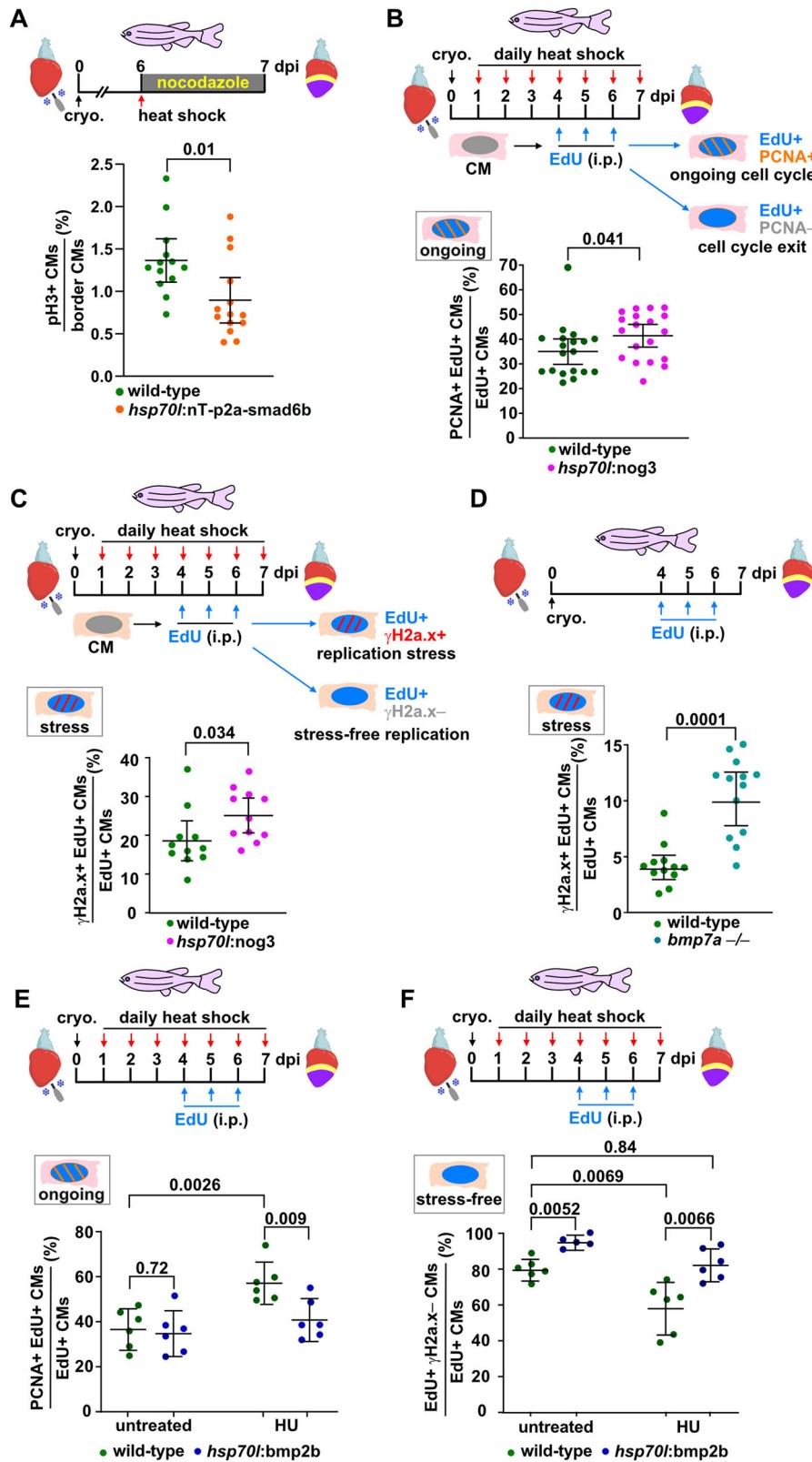

increased by 3.5% from an average ratio of 1.53 in controls to 1.59 in BMP2 treated cells), BMPs did not affect the frequency at which fork stalling occurred in either HSPCs or U2OS cells (Supplementary Fig. 10A, B). Yet, BMP signaling might alleviate replication stress by promoting re-start of stalled forks. To test this, we first allowed U2OS cells to incorporate CldU into newly synthesized DNA, halted these ongoing replication forks using treatment with HU, and identified re-

starting forks after HU-washout and IdU treatment as those where CldU tracks were followed by IdU tracks (Fig. 8C). Intriguingly, both BMP2 and BMP4 treatment significantly enhanced the fraction of forks that managed to re-start after HU-induced arrest (Fig. 8C). We conclude that BMP signaling is sufficient to enhance replication dynamics, and propose that it alleviates replication stress at least in part via its ability to enhance replication fork re-start.

**Fig. 6 | BMP signaling promotes stress-free replication and relieves CMs from cell cycle delay. A** BMP-Smad LOF using *hsp70l*:nT-p2a-smad6b reduces CM mitosis within 24 h after a single heat-shock. $n_E = 2$, $n_A = 13$ wt, 14 *hsp70l*:nT-p2a-smad6b, $n_C = 19377$ wt, 17328 *hsp70l*:nT-p2a-smad6b. **B** Cycling CMs were labeled by EdU incorporation from 4 to 6 dpi, followed by EdU and PCNA co-staining at 7 dpi. CMs that are EdU+ PCNA+ display an ongoing cell cycle. BMP-LOF using *hsp70l*:nog3 enhances the fraction of EdU+ PCNA+ CMs at 7 dpi over that observed in heat-shocked wild-type fish. $n_E = 2$, $n_A = 19$ wt, 19 *hsp70l*:nog3, $n_C = 14830$ wt, 14981 *hsp70l*:nog3. **C** Cycling CMs were labeled by EdU incorporation from 4–6 dpi, followed by EdU and γH2a.x co-staining at 7 dpi. Double-positive CMs are considered to experience replication stress, EdU+ γH2a.x– CMs are stress-free. *hsp70l*:nog3 enhances the fraction of EdU+ γH2a.x+ stressed CMs at 7 dpi. $n_E = 1$,

$n_A = 11$ per group, $n_C = 5440$ wt, 5660 *hsp70l*:nog3. **D** *bmp7a –/–* mutants display an increase in the fraction of CMs that experience replication stress at 7 dpi. $n_E = 2$, $n_A = 12$ wt, 13 *bmp7a –/–*, $n_C = 1500$ wt, 1300 *bmp7a –/–*. **E** HU treatment increases the fraction of EdU+ PCNA+ CMs at 7 dpi, which can be reversed by BMP-GOF using *hsp70l*:bmp2b. $n_E = 1$, $n_A = 6$ per group, $n_C = 26000$ CMs across all groups. **F** BMP-GOF using *hsp70l*:bmp2b is sufficient to increase the fraction of CMs that experience stress-free replication at 7 dpi, even when additional stress is induced by HU treatment. $n_E = 1$, $n_A = 6$ wt (untreated), 5 *hsp70l*:bmp2b (untreated), 6 wt (HU), 6 *hsp70l*:bmp2b (HU). $n_C = 18910$ CMs across all groups. **A–F** Data are presented as mean values. Error bars, CI 95%. Two tailed Student´s *t*-test. Source data are provided in the Source Data file. Some elements were created in Biorender (Agreement number: AH27UPOG7H; Posadas, (2025) https://BioRender.com/w88l548).

## Discussion

In summary, our data suggest that alleviation of replication stress is a conserved feature of BMP signaling, which is essential for the high capacity of adult fish to regenerate the heart. We propose the following model (Fig. 9): Cardiomyocyte (CM) regeneration via proliferation of spared CMs is highly efficient and capable of fully restoring pre-injury CM numbers in adult zebrafish. CMs are nevertheless not immune from challenges to DNA replication, which are thought to restrict tissue renewal and stem cell function in aged mammals. In the absence of BMP signaling or upon inhibition of DNA damage response pathways, cycling CMs that experience replication stress cannot overcome it, get stuck in S-phase of the cell cycle, and regeneration fails. In contrast to other pro-regenerative signaling pathways that promote CM cycling and thereby also increase replication stress, BMP signaling allows for stress-free replication by promoting re-start of stalled replication forks. This ability of BMP signaling is conserved also in mammalian cells; thus, activation of the BMP pathway might have potential as anti-aging and pro-regenerative intervention in aged individuals.

The concept that highly efficient regeneration depends on the ability to overcome challenges that limit tissue repair and homeostasis in aged mammals is also supported by work in salamanders. The Whited lab showed that blastema cells in regenerating axolotl limbs experience replication stress and that interference with DNA damage response pathways impairs limb regeneration[45]. Additionally, senescent cells accumulate in regenerating salamander limbs[46]. Yet, these are efficiently cleared and appear to actually promote regeneration[46,47]. We do not envision that regeneration-induced replication stress in the zebrafish heart has a beneficial function; rather we assume that it is an unavoidable by-product of the demands that regeneration puts on cell cycling. We propose that a major reason why BMP signaling promotes CM regeneration in zebrafish is because it allows the cells to alleviate replication stress.

The molecular reasons for regeneration-induced replication stress in axolotl blastema cells and zebrafish CMs remain to be identified[45]. Since we do not observe a regeneration induced increase in overall rates of transcription in border zone CMs, conflicts of replication forks with transcription are an unlikely cause of replication stress. Zebrafish CMs also do not appear to suffer from oxidative DNA damage, which limits CM proliferation in postnatal mice. Our data also suggest that γH2a.x accumulation is not due to pre-existing DNA lesions, which could interfere with progression of replication forks, since the fraction of γH2a.x+ CMs does not increase with age of the fish. Thus, the molecular basis of CM replication stress during regeneration warrants further investigation.

Our data show that ATR and ATM inhibition reduces CM proliferation. This supports our conclusion that CM replication stress needs to be sensed and its mitigation initiated by these kinases in order for CMs to be able to proceed from S-phase into mitosis. However, ATR is required for mammalian embryonic development and unstressed cells deficient of ATR can initiate DNA replication, but experience dNTP depletion, fork stalling and replication failure[48,49]. Thus, ATR can be required for proliferation of normal unstressed cells, in particular when

they enter the cell cycle from G0 phase and when cellular resources are limited. Zebrafish cardiomyocytes exhibit minimal cell cycle activity under homeostasis, and regeneration likely imposes significant demands on the cell cycle machinery, raising the possibility that ATR is also required for successful completion of S-phase in regenerating CMs that do not experience replication stress. While we cannot fully exclude this, we do not consider it likely, since we have used doses of the ATR inhibitor that did not perturb zebrafish embryonic development, and thus do not generally interfere with the proliferation of unstressed cells.

We found that BMP-Smad signaling is required for alleviation of CM replication stress in zebrafish regenerating hearts, and that it is sufficient to reduce replication stress in zebrafish and neonatal mouse CMs, as well as in human fibroblasts, U2OS cancer cells and HSPCs. Of note, pretreatment of cells with BMP ligands can very efficiently protect them from hydroxyurea-induced replication stress. Co-treatment (as in the case of mouse CMs) however also strongly reduces the stress, indicating that BMP signaling does not only increase the resilience of cells towards subsequently induced stress. Rather it likely acts more directly in the alleviation of the stress. Our DNA fiber spreading assays suggest that it does so, at least in part, via promotion of replication fork re-start.

Our findings on the role of BMP signaling in alleviation of replication stress align closely with a recently published study which highlights a crucial role of BMP in ensuring replication fidelity in pluripotent cells and during neurogenesis[50]. In these systems BMP signaling mitigates replication stress by modulating replication fork dynamics and it reduces chromosome missegregation. BMP signaling has also been shown to be protective in endothelial cells against mitomycin C mediated DNA damage[51]. In cultured cells with active BMP signaling, where Smad1 is phosphorylated at the SXS motif, genotoxic stress can, via the ATM kinase, result in additional phosphorylation of Smad1 at a different site[52]. The dually phosphorylated Smad1 can suppress degradation of p53 by Mdm2, resulting in p53 stabilization[52]. When focusing on p53´s canonical roles, these findings imply that active BMP signaling increases the sensitivity of cells to genotoxic stress and drives them towards cell cycle exit, apoptosis and senescence[52]. Yet, this role of BMP signaling appears to be quite different from how we propose it acts in regenerating zebrafish cardiomyocytes. Here, our results indicate that BMP signaling does not promote apoptosis or senescence. Instead, we hypothesize that it enhances progression of CMs that experience replication stress through the cell cycle. Future work will be necessary to clarify to which extent the different ways in which BMP signaling impinges on genomic stress are cell-type and context-specific.

Accumulation of γH2a.x in CMs of injured zebrafish hearts at 3 dpi has also been reported in a previous study, which however did not investigate its significance in wild-type hearts, but found that telomerase-dependent lengthening of telomeres in cardiac cells, including CMs, is essential for zebrafish heart regeneration[29]. In fish lacking *tert* telomerase activity, CM proliferation was reduced, γH2a.x accumulated in many more CMs, and CM senescence could be detected[29]. Thus, *tert* activity and BMP signaling as identified by our study represent the only pathways known so far that promote zebrafish heart regeneration by protecting CMs from genomic stress.

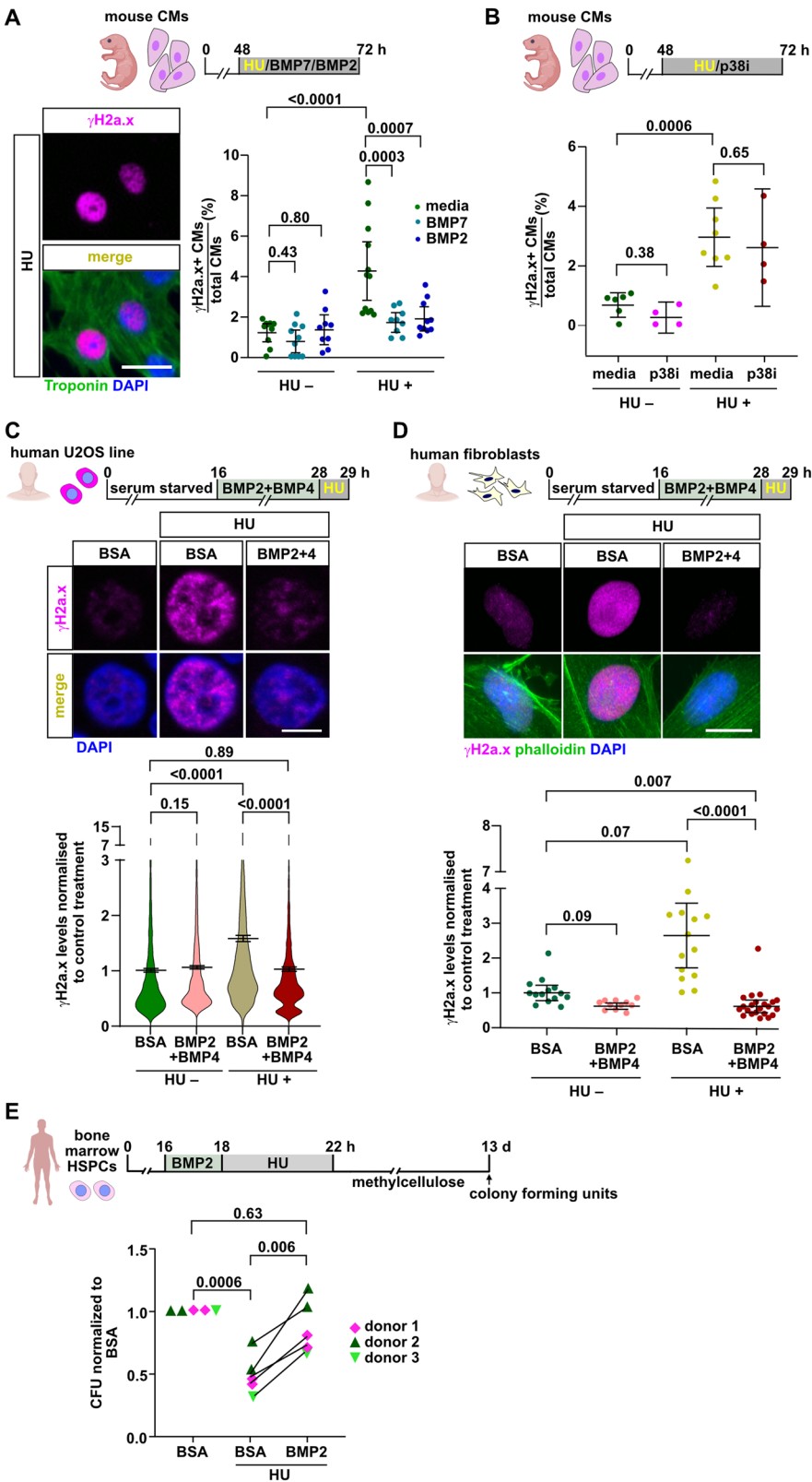

The emerging view that the elevated regenerative capabilities of certain species and organs, including the zebrafish heart, depends on highly efficient mechanisms to overcome cellular phenomena like replication stress that limit tissue renewal and repair in aged mammals, suggests that elucidation of such mechanisms could inform future anti-aging strategies.

## Methods

### Zebrafish husbandry and fish lines

All experiments involving zebrafish were approved by the state of Baden-Württemberg and the animal care representatives of Ulm University. Zebrafish were kept under standard conditions at 26–27 °C water temperature and a 14/10 h light/dark cycle. If possible, mixed

**Fig. 7 | The ability of BMP signaling to alleviate replication stress is conserved in mouse and human cells. A** Immunofluorescence for γH2a.x and Troponin shows that BMP2 or BMP7 rescue cultured primary neonatal mouse CMs from HU-induced replication stress. Data points represent the fraction of γH2a.x+ CMs per well. $n_E = 2$, $n_{wells} = 10$ media, 10 BMP7, 9 BMP2, 12 media (HU), 9 BMP7 (HU), 10 BMP2 (HU). $n_C = 2795$ media, 2658 BMP7, 2106 BMP2, 3593 media (HU), 2704 BMP7 (HU), 2533 BMP2 (HU). Scale bar, 20 µm. **B** The p38 MAPK inhibitor SB202190 does not protect CMs from HU-induced replication stress. $n_E = 2$, $n_{wells} = 6$ media, 4 p38i, 8 media (HU), 4 p38i (HU), $n_C = 1384$ media, 765 p38i, 1937 media (HU), 876 p38i (HU). The observed relative difference between HU (media) and HU (p38i) groups is 11%, the calculated smallest significant difference 36%, which is smaller than the effect size between the HU (media) and HU (BMP7) groups of 44% (from Fig. 7A). We conclude that this experiment had enough power to detect effects of that magnitude. **C** Immunofluorescence for γH2a.x shows that pretreatment with BMP2 and BMP4 protects U2OS cells from HU-induced replication stress. Plots show integrated intensity of γH2a.x nuclear levels. $n_E = 1$, $n_{wells} = 3$ for each treatment. Scale bar, 20 µm. **D** Immunofluorescence for γH2a.x and Phalloidin in human primary neonatal foreskin dermal fibroblasts shows that pre-treatment with BMP2 and BMP4 alleviates HU-induced replication stress. Data points represent quantification of γH2a.x nuclear levels. $n_E = 1$, $n_C = 14$ BSA, 11 BMP2 + 4, 14 BSA (HU), 22 BMP2 + 4 (HU). Scale bar, 20 µm. **E** Pre-treatment with BMP2 rescues survival and proliferative capacity of primary human bone-marrow derived HSPCs treated with HU. Data represent the colony forming units observed in HSPCs isolated from 3 individual donors normalized to BSA-treated controls. $n_E = 5$, $n_{donors} = 3$, $n_{plates} = 2$ for donors 1 and 2; 1 for donor 3. **A**–**C**, **E** Ordinary one-way ANOVA with Bonferroni correction; **D** Kruskal-Wallis followed by Dunn´s correction. **A**–**E** Data are presented as mean values. Error bars, CI 95%. Source data are provided in the Source Data file. Some elements were created in Biorender (Agreement number: CH27UPPBB6; Posadas, (2025) https://BioRender.com/n30v908).

groups of males and females were used for all experiments. Since significantly different results between males and females were not observed in any of the experiments, data from both sexes are reported together. The following previously published transgenic or mutant fish lines were used: *hsp70l*:nog3[fr14tg53]; *hsp70l*:bmp2b[fr13tg53]; *myl7*:GFP[twu34Tg54]; *14.8gata4*:GFP[ae155]; *myl7*:H2b-GFP[zf521Tg56]; *hsp70l*:bmp7b,*myl7*:eGFP[afsTg36]; *hsp70l*:bmp4,*myl7*:eGFP[afTg57]; *bmp7a*[ty68a/ty68a58]. Transgenic or mutant fish were raised together with their wild-type siblings to adulthood; fish were genotyped and separated only shortly prior to any experiment, and wild-type siblings from the same cross were used as negative controls in all experiments.

### Genotyping
*hsp70l*:bmp2b[fr13tg] and *hsp70l*:nog3[fr14tg] fish were genotyped by PCR. Genomic DNA was isolated from fin clips using 50 mM NaOH, followed by incubation at 95 °C for 20 min or until tissue was friable. Next, 1/10th volume of 1 M Tris-HCl, pH 8.0 was added. The following primers were used (the *hsp70l* forward primer is located in the *hsp70l* promoter and works for both transgenes and is combined with transgene-specific reverse primers: *Hsp70l* (ZDB-GENE-050321-1) GTGGACTGCC-TATGTTCATCTTATTTAGGTCTAC; *bmp2b* (ZDB-GENE-980526-474) ACACCTGACCGAGCAACAGC; *noggin3* (ZDB-GENE-990714-8) GTGGCCAGGAAATACGGTATGTTATCCAT. PCR amplification was conducted according to the program: initial denaturation at 94 °C for 3 min, followed by 35 cycles consisting of denaturation at 94 °C for 30 sec, annealing at 58 °C for 30 sec, extension at 68 °C for 30 sec and final extension at 68 °C for 2 min.

### Creation of the *hsp70l*:nT-p2a-Smad6b[ulm16Tg] transgenic line
The following DNA elements were assembled by PCR- and Gibson-assembly-based cloning methods in a vector containing a single I-SceI meganuclease restriction site: zebrafish *hsp70-4* 1.5 kb promoter, nuclear localization signal, dTomato, p2a "ribosome skipping" peptide, zebrafish *smad6b* coding sequence (ZFIN gene ID ZDB-GENE-050419-198), SV40 late polyadenylation site. I-SceI mediated transgene insertion was used to create a stable transgenic line. One subline was selected based on its ability to mimic known BMP loss-of-function phenotypes after heat-shock during gastrulation (dorsalization) and widespread expression after heat-shock in adult hearts.

### Rescue of zebrafish *bmp7a* (*snailhouse*) mutants and KASP genotyping
For adult experiments involving *bmp7a* homozygous mutants, embryos from incrosses of *bmp7a (snh)*[ty68a] heterozygous carriers were injected with 300 pg of *bmp7a* mRNA. After raising them to adulthood, genomic DNA was extracted from fin biopsies, diluted 1:6 in nuclease-free H$_2$O and subjected to Kompetitive allele specific PCR (KASP) genotyping. The 2X KASP master mix and KASP assay primer mix were designed by LGC Biosearch Technology from the following *bmp7a* CDS sequence

(TACTCTTATGAACCCGCGTACACGACCCCGGGACCCCCGCTGGTGACCCAGCAGGACAGTCGCTTTCTCAGTGATGCCGACATGG**[T/G]**GATGAGCTTTGCGAATACAGGTGAGCGTCTTATGAAATTCACCGCATATCATAATTGTTGTTAGGATGAATCAACAGATTGTTTTTGCTCCATT). The cycling program for KASP PCR involved a hot start activation at 94 °C for 15 min, 10 cycles at 94 °C for 20 sec and at 61 °C for 60 sec, 26 cycles at 94 °C for 20 sec and at 55 °C for 60 sec, followed by 3 cycles at 94 °C for 20 sec and at 57 °C for 60 sec. FAM fluorophore (T) corresponds to the wild-type allele and the HEX fluorophore (G) corresponds to the mutated one. The expected genotype distribution is 25% wild-type, 50% heterozygous and 25% homozygous mutants. Adult tanks contained an average of 24% homozygous mutant fish.

### Zebrafish heart cryoinjury, resection and heat-shock-based transgene expression
Ventricular resections were performed as previously described[1]. Cryoinjuries were performed as previously described[5] except that a liquid nitrogen-cooled copper filament of 0.3-mm diameter was used instead of dry ice. For most experiments, heart injuries were performed on zebrafish of 4–6 months of age. For the aging experiments, zebrafish of 24 months of age were used. For juvenile experiments, fish of 8–9 weeks of age and 11–16 mm standardized standard length were used. Injured fish were kept at a density of 7 fish per 1.5 liters. Efforts were made to keep fish numbers equal between experimental groups. In case of death or when equal numbers were unavailable fish from a different, easily identifiable wild-type background were added to equalize the numbers. Heat-shocks were applied by heating water containing fish from 27 °C to 37 °C within 10 min, and reducing temperature after 1 h back to 27 °C within 15 min. For long-term experiments, fish were heat-shocked once daily for 7 days; for short-term experiments fish were heat-shocked once at 7 dpi and hearts were harvested 5 h after the end of the heat-shock.

### Zebrafish growth manipulations
Juvenile fish at an age of 8–9 weeks (11–16 mm standardized standard length) were either maintained at a high density of 40 fish / liter for a period of 10 days (restricted growth) or a low density of 2 fish / 11 liter (stimulated growth). For starvation experiments, control fish were fed normally (a combination of artemia and flake food 2–3 times a day) while starved fish were fed with artemia once daily every 2nd day. Fish were starved starting 3 days prior to injury throughout the entire duration of the experiment.

### Pharmacological interventions
Detailed information on drugs used is listed in Supplementary Table 1. For EdU experiments, adult fish were injected intraperitoneally with 10 µL of 10 mM 5-Ethynyl-2′-deoxyuridine diluted in PBS. Juvenile fish were soaked in EdU (1 mM) dissolved in 50 mL E3 medium (5 mM NaCl,

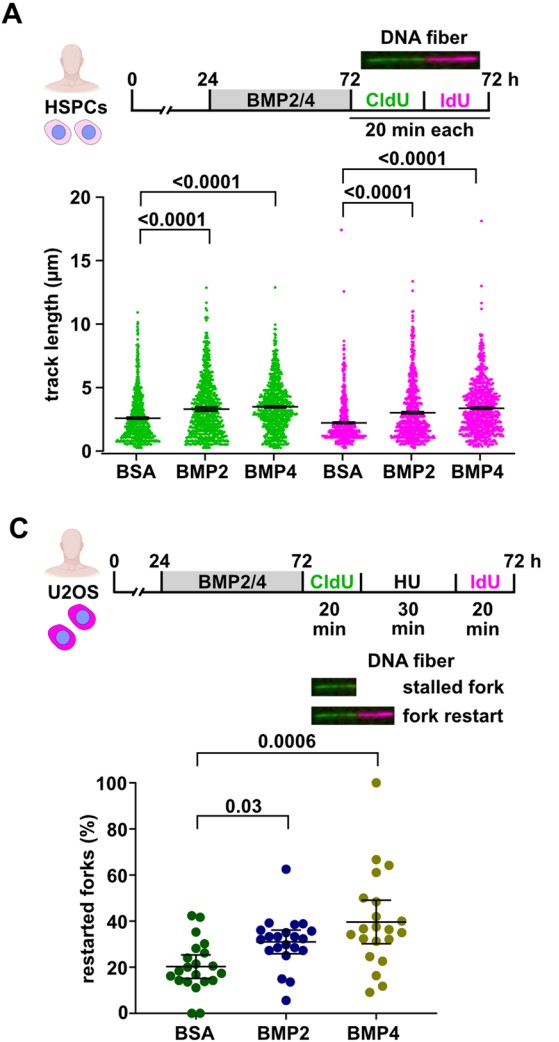

**Fig. 8 | BMP signaling resolves replication stress by enhancing replication fork speed and re-start. A** DNA fiber spreading assays show that treatment with either BMP2 or BMP4 ligands for 48 h increases the speed of replication fork progression measured by track length of either CldU (green) or IdU (red) in human cord blood HSPCs. Data points represent length of individual CldU or IdU tracks. $n_E = 3$, $n_{fibers} = 670$ per treatment. **B** Treatment with BMP4 increases replication fork progression in U2OS cells, while BMP2 does not have a significant effect. $n_E = 2$, $n_{fibers} = 250$ per treatment. **C** Pre-treatment with BMP2 or BMP4 ligands increases the fraction of replication forks that restart after HU-mediated stalling in U2OS cells. Fork arrest is indicated by tracks that are only labeled by CldU, restarted forks by CldU tracks that are followed by IdU. Data points represent fraction of restarted forks out of all analyzed forks. $n_E = 2$, $n_{fibers} = 600$ per treatment. **A–C** Data are presented as mean values. Error bars, CI 95%. Kruskal-Wallis followed by Dunn´s correction. Source data are provided in the Source Data file. Some elements were created in Biorender (Agreement number: CH27UPPBB6; Posadas, (2025) https://BioRender.com/n30v908).

0.17 mM KCl, 0.33 mM CaCl$_2$ x 2H$_2$O, 0.33 mM, MgSO$_4$ x 7H$_2$O, 0.2% (w/v) methylene blue, pH 6.5). For detection of mitotic cardiomyocytes, control and experimental fish were soaked with 5 μM nocodazole added to 600 mL of fish facility system water to arrest cells in mitosis. Adult fish were treated with hydroxyurea (20 mM), Rapamycin (1 μM), ATM kinase inhibitor KU55933 (1 μM), ATR inhibitor VE-821 (1 μM), Cdk4/6 inhibitor PD0332991 (2 μM) by soaking the fish with respective concentrations in 1 L of fish water. DMSO was used to dissolve Rapamycin, ATM, ATR kinase and Cdk4/6 inhibitors. Calcidiol was applied by intraperitoneal injection of 10 μL of 200 μM α-calcidiol dissolved in 10% ethanol.

## Microarray analysis and GSEA protocol

Microarray-based transcriptomics of zebrafish heart regeneration was performed on an Agilent platform with custom-designed zebrafish probes (Supplementary data file 1). Ventricular resection was used as injury model, and RNA was isolated from whole ventricles. Uninjured hearts were compared with regenerating hearts at 2, 4 and 7 dpi timepoints. Differentially regulated genes between uninjured and 7 dpi

groups were identified using the Bioconductor package Limma (Linear Models for Microarray Data, Supplementary data file 1). For these DEGs, GO enrichment analysis was performed using R package clusterProfiler[59] against the Genome wide annotation for Zebrafish (R package org.-Dr.eg.db, DOI: 10.18129/B9.bioc.org.Dr.eg.db) To find interesting functional terms, the analysis was conducted using the biological process category of the GO, with the following parameters: annotation cutoff of min 10 and max 500 counts, and a *p*-value cutoff of 0.05 (Supplementary data file 2). The terms with FDR adjusted (*P*.adjust) value < 0.05 were considered as significantly enriched. The enrichment score plots were generated using the R package enrichplot (https://bioconductor.org/packages/release/bioc/html/enrichplot.html).

## Mining of published single cell RNA sequencing data

Zebrafish regenerating heart single-cell RNA-seq data was taken from the Gene Expression Omnibus (GEO) database under the accession number GSE251856. Clusters representing CMs were identified based on expression (or enrichment) of *tnnt2a, ttn.1, ttn.2, myl7*, and *pln*. Border zone CMs within the CM cluster were based on the presence/

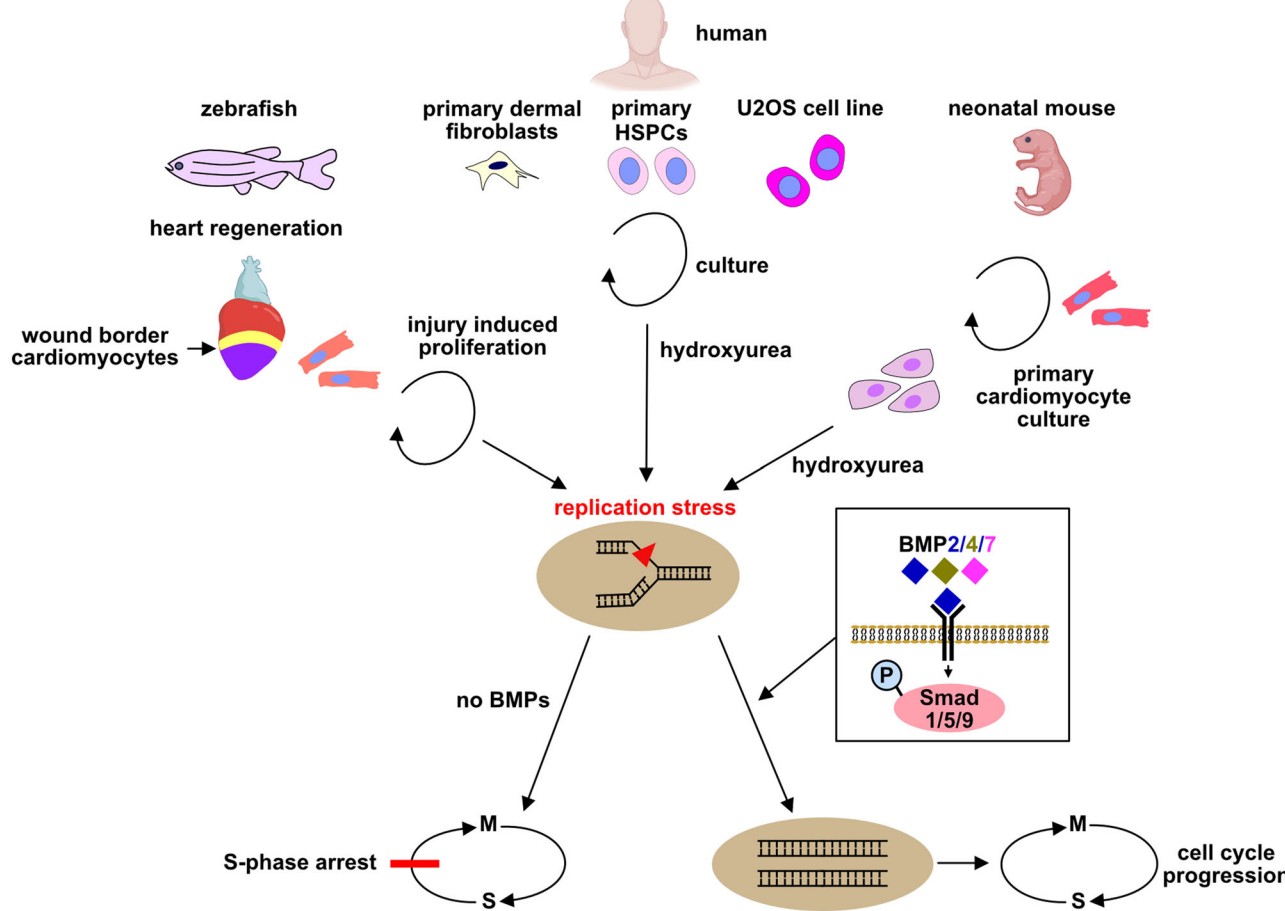

**Fig. 9 | Model for the role of BMP signaling in alleviation of replication stress in zebrafish heart regeneration and mammalian cells.** Zebrafish hearts efficiently regenerate via proliferation of pre-existing cardiomyocytes, although cardiomyocytes experience regeneration-induced replication stress. BMP signaling, activated by BMP2, 4 and 7 ligands, and mediated via Smad-signaling, allows CMs to overcome the replication stress, which otherwise leads to S-phase arrest and failure or regeneration. The unique ability of BMP signaling to promote stress-free replication is conserved in mouse cardiomyocytes, human cell lines, primary fibroblasts and hematopoietic stem and progenitor cells. Some elements were created in BioRender (Agreement numbers: AH27UPOG7H; https://BioRender.com/w88l548; CH27UPPBB6; Posadas, D. (2025) https://BioRender.com/n30v908).

enrichment of the following transcripts[11,60]: *mustn1b*, *tagln*, *myl6*. Differential gene expression analysis (DEG) was performed using Scanpy.

### qRT-PCR
For qRT-PCR on regenerating zebrafish hearts, RNA was isolated from whole ventricles using the RNeasy for fibrous tissue kit (Qiagen) following the manufacturer's instructions. cDNA was synthesised using the LunaScript RT SuperMix Kit (New England Biolabs, E3010L) as per manufacturer's protocol. Reactions containing no reverse transcriptase were used to exclude genomic DNA contamination. qRT-PCR was performed using the Luna Universal qPCR Master Mix (NEB, M3003L) in a BioRad machine. Ct values were normalised to the geometric mean of three housekeeping genes *18S rRNA* (ZDB-RRNAG-180607-2), *ubb* (ZDB-GENE-050411-10) and *Eef1a1l* (ZDB-GENE-990415-52). Differential expression was calculated using the $-2^{\wedge\Delta\Delta Ct}$ method[61]. Primer amplification efficacy was determined on serial 1:4 dilutions of cDNA mix prepared from wounded hearts using the same conditions as those used for experimental samples. Primers were only used if their amplification efficacy exceeded 2. Primer sequences are listed in Supplementary Table 2.

### Western blot of zebrafish heart wound border zones
The wound was identified in extracted whole hearts using stereomicroscopy based on its distinctive color relative to unaffected healthy myocardium. Next, it was removed using fine forceps and scissors. The basal part of the ventricle was removed as well, leaving the border zone for homogenization. Border zones from 10 hearts were pooled per sample. Border zone tissues were washed with PBS and manually homogenized in 100 µl SDS lysis buffer (63 mM Tris-HCl, 10% glycerol, 5% 2-mercaptoethanol, 3.5% SDS) using Wheatons glass homogenizers in 1 mL plastic tubes until no particles were seen, followed by centrifugation to clear lysates. SDS-PAGE and immunoblots were performed following standard procedures with a Bio-Rad system. Blotting was conducted onto nitrocellulose membranes, which were incubated overnight with primary antibodies. Information about primary antibodies is listed in Supplementary Table 3. After three washes 10 min each with TBS (20 mM Tris-HCl, 500 mM NaCl, pH 7.4) Li-cor secondary antibodies were utilized for visualization (Supplementary Table 4), and quantification was performed within a linear range of exposure using a Li-COR ODYSSEY Imager and Image Studio Light software. Intensities of p-Rpa32 and γH2a.x protein bands were normalised to Gapdh signals from the same lysate and blot.

### γ-Irradiation
Zebrafish were irradiated using a Cs-137 source at 3 Gy / min. Adults were irradiated with 40 Gy. Embryos were irradiated with 8 Gy.

### DNA damage competitive ELISA
Wild-type hearts were harvested at 7 dpi and cultured ex vivo following published protocols[62]. As positive controls, uninjured and injured

hearts were incubated in 100 mM $H_2O_2$ for 1 h at 28 °C, 5% $CO_2$. DNA was extracted using QIAGEN DNeasy Blood & Tissue Kit (Cat #69504) as per manufacturer´s protocol. Extracted DNA was quantified using Nanodrop, denatured at 95 °C for 10 min and rapidly chilled on ice to produce single-stranded DNA. The samples were digested with 10 units of Nuclease P1 per 1 µg of DNA for 2 h at 37 °C, followed by treatment with 5 units of alkaline phosphatase in 100 mM Tris (pH 7.5) at 37 °C for 1 h[63]. Supernatant was used to measure 8-OHdG levels using a DNA Damage Competitive ELISA Kit (ThermoFisher Cat# EIADNAD). The 8-OHdG levels were within the range of the standard curve.

### Tissue processing and immunostaining

Zebrafish hearts were fixed in 4% paraformaldehyde (PFA) in phosphate buffer at RT for 1 h. For γH2a.x stainings hearts were fixed for 2 h. Subsequently, they were washed three times for 10 min in 4% sucrose/phosphate buffer and equilibrated in 30% sucrose/phosphate buffer overnight at 4 °C. Hearts were embedded in Peel-A-Way Embedding Molds (Sigma Cat#18646A-1) using NEG-50 frozen (ThermoFisher Cat #1214849) section medium and cryosectioned into 10 µm sections. These sections were evenly distributed onto six serial slides to ensure representation from all ventricular areas on each slide. For immunostainings, slides were washed three times with PEMTx buffer (80 mM Na-PIPES, 5 mM EGTA, 1 mM $MgCl_2$ pH 7.4, 0.2% Triton-100) and once with PEMTx + 50 mM $NH_4Cl$. Slides were blocked in PEMTx/NGS (10% Normal Goat Serum, 1% DMSO, 89% PEMTx) for 1 h at RT in a humidified chamber. The primary antibodies were applied in PEMTx/NGS at 4 °C overnight. For PCNA, antigen retrieval was performed by incubating the slides in pre-heated 10 mM sodium citrate at 95 °C in a water bath for 10 min. The primary and secondary antibodies used are listed in Supplementary Tables 3 and 4, respectively. Secondary antibodies were used at a dilution of 1:1000. Nuclei were shown by DAPI (40,60-diamidino-2-phenylindole) staining. Slides were mounted with Fluorsave (Merck Cat #345789) mounting medium. For EdU detection, cryosections were prepared the same manner as for immunofluorescence staining. Slides were washed twice in 3% BSA/PBS for 5 min, once in 0.5% Triton X-100/PBS for 20 min, and twice in 3% BSA/PBS for 3 min. Next, the slides were incubated in the dark with the reaction cocktail (EdU-Imaging kit, baseclick GmbH) as per manufacturer's protocol for 30 min. Finally, slides were washed for 3 min in 3% BSA/PBS, followed by the immunofluorescence protocol. For β-Galactosidase staining, Senescence β-Galactosidase Staining Kit (Cell Signaling #9860) was used as per manufacturer´s protocol. Hearts were fixed with 1X Fixative solution (#11674) for 15 min at RT. Hearts were washed twice with 1X PBS, after which β-Galactosidase Staining solution (1X staining solution #11675, 100X solution A #11676, 100X solution B #11677, and X-gal #11678) was added and incubated overnight at 37 °C. Stained hearts were then cryosectioned. For AFOG staining, cryosections were heated in Bouin´s solution at 60 °C for 2 h, followed by an additional h at RT. The slides were then rinsed under slow-flow running water for 30 min until the water ran clear. Next, slides were incubated in 1% Phosphomolybdic acid for 5 min followed by a 5 min rinse in $ddH_2O$. Slides were incubated in AFOG solution: aniline blue (Sigma #415049), orange G (Fluka 7#3580) and acid fuchsin (Sigma #F8129) for 5 min, followed by three 5 min rinses in $ddH_2O$, dehydration twice in 95% ethanol and twice in 100% ethanol. Finally, slides were placed in xylene twice for 2 min and mounted with BioCare EcoMount (ACDBio #320409).

### Image acquisition and analysis

All immunofluorescence images were acquired as single optical planes using 20 x magnification (unless otherwise specified) either with Leica Sp5/Sp8 confocal microscopes or with a Zeiss AxioObserver 7 equipped with an Apotome. All image quantifications were performed using ImageJ standard functions. Quantifications of the fraction of EdU, PCNA, γH2a.x and p-Smad positive cardiomyocytes were performed manually on 2 to 3 sections per heart that contained the biggest wounds and were restricted to the myocardium located within 150 µm from the wound border. To define this region of interest, the wound border was identified based on immunofluorescence for cardiomyocyte markers. The wound was selected in ImageJ using the "free hand selection" tool and added to the Region of Interest (ROI) manager. Next the selection was enlarged by 150 µM and added to the ROI manager. Both ROIs were selected and XOR function with overlays was performed. In some cases, especially where pH3 was quantified, the number of cardiomyocytes in the border zone was estimated by multiplying the wound border area (measured using ImageJ) with the average density of cardiomyocytes in three separate regions of interest (size: 100 µm²) determined by manual counting of cardiomyocyte nuclei within the regions of interest. For each heart the average value was calculated from the analysis of 2–3 sections per heart. In all other cases, the number of all cardiomyocyte nuclei in the ROI was counted manually for each section separately. For cardiomyocyte mitotic index estimation, the number of pH3+ cardiomyocytes was calculated from the analysis of 8 to 10 sections per heart. For measuring p-Pol2 (S5) intensity, CM nuclei were identified using GFP signal in *myl7*:H2b-GFP transgenic hearts with the "Particle Analysis" tool in Image J. A selection (ROI1) was created for these nuclei after setting a threshold to remove background. The wound border zone was identified as described above and a separate selection was created (ROI2). Using the AND function in ImageJ, ROI1 and ROI2 were combined to generate a selection for nuclei of the border zone resulting in ROI3. In the p-Pol2 (S5) channel, background removal was performed and ROI3 was applied. Intensity was measured using the "Measure" command for individual nuclei. Measurements of the wound size area and of the entire ventricle area were performed manually using ImageJ on all sections of one serial slide representing 1/6 of the total ventricle. For collagen measurements, the blue area was first separated with the "Threshold Color" tool using the "Hue" function and then measured manually with ImageJ.

### Mouse cardiomyocyte cultures, drug treatments and immunofluorescence

Postnatal day 0 or 1 (P0-P1) cardiomyocytes were isolated from wild-type C57BL/6 mouse hearts by mechanical digestion, using a magnetic stirrer, and enzymatic digestion in ADS (5 mM glucose, 106 mM NaCl, 5.3 mM KCl, 20 mM Hepes, 0.8 mM $Na_2HPO_4$, and 0.4 mM $MgSO_4$, pH7.4) with 1 mg/ml pancreatin (Sigma #P1750) and 0.45 mg/ml collagenase A (Roche #10103586001), as previously described in ref. [64]. Eight to ten hearts were used for each digestion which yielded around $1 \times 10^6$ cells. 35,000 were seeded per well in 96 well-plates. Cardiac cells were then cultured and allowed to adhere for 48 h in 0.1% gelatine-precoated (Sigma #G9391) plates, with DMEM/F12 (Thermofisher), 1% L-glutamine (Sigma #G7513), 1% sodium pyruvate (Life Technologies #11360-039), 1% non-essential amino acids (Life Technologies #11140-035), 1% penicillin and streptomycin (Gibco #15140122), 5% horse serum (Invitrogen #16050-122) and 10% FBS (Gibco #10082147) and maintained in a humidified atmosphere (5% $CO_2$) at 37 °C. Thereafter, for replication stress analysis, the medium was replaced with medium containing 5% horse serum without FBS to limit fibroblast growth, supplemented with 0.5 mM Hydroxyurea (Sigma) and the following treatments: BMP7 (R&D #5666-BM-010), BMP2 (R&D #355-BM-010), or p38 MAPK inhibitor SB202190 (Sigma #S7067) at 10 µM, for 24 h.

Cardiac cells were fixed with cold 4% paraformaldehyde (PFA) solution for 20 min at RT and then washed three times in PBS and once in 3% BSA (Sigma) in PBS for 5 min. Cells were permeabilized with 0.5% Triton-X100 in PBS for 5 min at RT and a blocking solution (PBS supplemented with 5% BSA (Sigma) and 0.1% Triton-X100) was applied for 1 h at RT. Then the cells were incubated overnight at 4 °C with primary antibodies diluted in PBS, supplemented with 3% BSA and 0.1%

Triton-X100 (see Supplementary Tables 3 and 4). Samples were washed thrice with PBS and incubated with fluorescent cross-adsorbed secondary antibodies (diluted 1:500 in PBS with 1% BSA (Sigma) and 0.1% Triton-X100 (Sigma)) conjugated to Alexa Fluor™ 555 and 488. After 3 washes in PBS, DAPI (40,60-diamidino-2-phenylindole) was applied for 10 min at RT. Samples were then washed two times in PBS and imaged at 20x magnification.

## U2OS cell culture, treatment and immunofluorescence
U2OS cells were cultivated in DMEM high glucose (Gibco by Thermo Fisher Scientific) supplemented with 10% FBS (Biochrom by Merck) at 37 °C, 5% $CO_2$ and ambient $O_2$. For BMP stimulation cells were transferred into serum-free medium X-VIVO 10 (Lonza). After 24 h in the serum-free medium, cells were treated with 50 ng/ml BMP2 or BMP4 (R&D Systems #355-BM-010 and #314-BP-010) solved in 0.1% BSA (Sigma-Aldrich) for 48 h. For γH2A.x staining, U2OS cells were serum starved for 16 h and then treated with BMP2 and BMP4 or BSA at a concentration of 50 ng/mL of each ligand for 12 h. Then they were treated with HU (Sigma, #H8627) at a concentration of 100 μM for 1 h. Following treatment, the cells were washed with PBS 3 times, fixed in 4% PFA for 10 min at RT. Immunofluorescence was performed by blocking with 3% BSA in PBS for 1 h at RT, overnight incubation at 4 °C with anti-γH2A.x antibody (Supplementary Table 3) 1:200. Three washes were then performed with PBS at RT, and incubation with 1:1000 Alexa flour 555 labeled secondary antibody for 1 h at RT in the dark followed by three washes in PBS. Nuclei were counterstained with DAPI (40,60-diamidino-2-phenylindole) for 5 min at RT. After 3 washes with PBS, slides were mounted in Dako Fluorescence Mounting Medium (Agilent, #GM304) and imaged at 40x magnification. γH2A.x intensity measurement was performed using CellProfiler[65].

## Human primary fibroblast cell culture, treatments and immunofluorescence
Human foreskin dermal fibroblasts were cultured in DMEM (Thermo Fisher, Cat No: 11965092) with 10% FCS and 1% Pen/Strep at 37°C and 5% $CO_2$. Fibroblasts were cultured up to 70–80% confluency, afterwards, fibroblasts were passaged 1:3 using accutase. For the experiments, fibroblasts were seeded at ~5000 cells/chamber in poly-D-lysin coated 4-chambered slides (Corning, #354577) in 1 ml of DMEM. Following attachment, fibroblasts were serum starved for 16 h and then treated with BMP2 and BMP4 (R&D Systems #355-BM-010 and #314-BP-010) or BSA at a concentration of 50 ng/mL of each ligand for 12 h. The fibroblasts were then treated with hydroxyurea (Sigma, #H8627) at a concentration of 100 μM for 1 h. Following treatment, the cells were washed with PBS 3 times, fixed in 4% PFA for 10 min at RT, and permeabilized in 0.2% Triton X-100 in PBS for 10 min at RT. Immunofluorescence was performed by blocking with 5% BSA in PBS for 1 h at RT, overnight incubation at 4 °C with anti-γH2A.x antibody (Supplementary Table 3) diluted 1:200 in Dako antibody diluent (Agilent, #S0809), three washes with PBS at RT, incubation with 1:400 Alexa flour 555 labeled secondary antibody for 1 h at RT in the dark, three washes in PBS, and staining with phalloidin-Alexa flour 488 for 30 min. After 3 washes with PBS, nuclei were counterstained with DAPI (40,60-diamidino-2-phenylindole) for 5 min at RT. After 3 washes with PBS, slides were mounted in Dako Fluorescence Mounting Medium (Agilent,#GM304) and imaged at 40x magnification.

## Cord blood HSPC cell culture
Human CD34+ hematopoietic stem and progenitor cells (HSPCs) were isolated from cord blood samples with consent of the parents (approval #155/13 from the Ethical Board of Ulm University) and were cultured as described in ref. 43 using the following procedure: Cord blood samples were diluted 1:2 in PBS and layered on Ficoll® Paque Plus (Sigma-Aldrich by Merck,) to isolate mononuclear cells. The cell layer was then transferred to a new tube and washed with PBS containing 2%

FBS. Red blood cells were removed by incubation in red blood cell lysis buffer (0.15 M ammonium chloride, 10 mM potassium-bicarbonate, 0.1 mM EDTA in water, pH 7.2–7.4), followed by another wash in PBS. CD34+ HSPCs were enriched using the CD34 Micro Bead Kit (Miltenyi Biotec) according to the manufacturer´s instructions. HSPCs were cultured for 24 h at 37 °C, 5% $CO_2$ and ambient $O_2$ in serum free StemSpanTM SFEM medium (STEMCELL Technologies) supplemented with 1% Penicillin/Streptomycin (Gibco by Thermo Fisher Scientific) and 1% StemSpanTM CC100 (STEMCELL Technologies,) before treatment with either 50 ng/ml BMP2 or BMP4 (R&D Systems #355-BM-010 and #314-BP-010) solved in 0.1% BSA (Sigma-Aldrich by Merck) for 48 h.

## DNA fiber spreading assay
DNA fiber spreading assays were performed as described before[66–68]. In brief, cells were labeled with 20 mM 5-chloro-2-deoxyuridine (CldU, Sigma-Aldrich by Merck) for 20 min, then washed with pre-warmed PBS, followed by a second labeling with 200 mM 5-iodo-2-deoxyuridine (IdU, Sigma-Aldrich). For HU treatment, cells were incubated after the first labeling with 5 mM HU for 30 min, then washed again with PBS before the second labeling. Cells were then washed, harvested and resuspended. In case of U2OS, cells were subsequently trypsinized and resuspended. Then 2500 cells were spotted on a slide and lysed with 6 μl lysis buffer (0.5% SDS, 100 mM Tris–HCl pH 7.4, 50 mM EDTA) for 6 min at RT. Slides were then tilted ~20° to allow the spreading of DNA via gravity. After further 6 min drying, DNA was fixed on the slides by incubation for 5 min with methanol-acetic acid solution (3:1). Slides were either dried for another 7 min and stored in 70% (v/v) ethanol at 4 °C overnight or directly subjected to denaturation/deproteination in 2.5 M HCl for 1 h, followed by immunofluorescence staining. Blocking with 5% (w/v) BSA in PBS for 45 min at 37 °C was followed by incubation with a primary antibody mix of anti-BrdU detecting IdU (mouse, BD #347580,) and anti-BrdU detecting CldU (rat, monoclonal, clone BU1/75 (ICR1), BioRad, #OBT0030) for 1 h at RT. Finally, slides were incubated with a mix of secondary antibodies, AlexaFluor555 (anti-mouse) and AlexaFluor488 (anti-rat, both secondary antibodies were from Invitrogen by Thermo Fisher Scientific,) for 45 min at RT. DNA fibers were visualized with a BZ-9000 microscope (Keyence) at 40x. Measurement of the DNA fiber track length was carried out with the Fiji software.

## Human bone-marrow HSPC colony forming assay
Human Bone Marrow CD34+ Progenitor Cells Cryopreserved (Lonza #2M-101) from distinct adult donors (sex/gender of the donor was not considered) were thawed and then diluted in 10 mL thawing buffer (IMDM with 20% FBS, 1% PenStrep, 400 I.E./mL Heparin, 0.01 mg/mL DNAse I). Cells were centrifuged at 300 g at 4 °C for 5 min and washed once with 5 mL working buffer (PBS + 3% FBS). Cells were seeded in StemSpan SFEM (StemCell Technologies #09650) with 100 ng/ml Stem Cell Factor (SCF; StemCell Technologies #78062,), 50 ng/ml Thrombopoietin (TPO; StemCell Technologies #78210) and 100 ng/ml Tyrosine Kinase 3 (Flt3; StemCell Technologies #78009) into a 96-well plate, pre-coated with 50 μg/mL of fibronectin. Cells were then incubated for 16–20 h in 3% $O_2$. After the initial incubation, cells were treated for 2 h with 0.1% BSA as control or 50 ng/mL of either BMP-2 or BMP-4 (R&D Systems #355-BM-010 and #314-BP-010), followed by treatment with 10 mM HU for 4 h. Cells were centrifuged at 300 g at 4 °C for 5 min and resuspended in 50 μL of SFEM. The cell mixture was then thoroughly mixed with 1.5 mL of previously thawed, room temperature methylcellulose (R&D Systems #HSC003). The methylcellulose was plated on a 35 × 10 mm dish with a grid (Thermo Fisher Scientific #174926) using a 18 G blunt end needle (VWR #613-2948). Cells were incubated for 12 days at 37 °C and 3% $O_2$. At day 12, the frequency of colony forming units (CFU) was determined. Samples that showed a physiological level of HU-induced stress (reduction of

CFU counts in the range from 80% – 20% of controls) were further analyzed for the ability of BMP2 to attenuate this HU-induced stress.

## Statistics

Statistical analyses were performed using GraphPad Prism 9 software. Information about samples sizes and types of statistical tests can be found in figure legends. Data are presented as mean values. Error bars represent CI 95%. If not stated differently, sample sizes are given in the figure legends as follows: $n_E$ = number of independent experiments or biological replicates, $n_A$ = animals (combined for all experiments/replicates), $n_C$ = number of cells analyzed (combined for all experiments/replicates). For performing statistical tests on fractions, data were transformed by computing their square root before conducting Two tailed Student´s $t$-test or Ordinary one-way ANOVA followed by Bonferroni correction. For individual measurements which were not normally distributed, either Mann–Whitney (2 groups) or Kruskal-Wallis test followed by Dunn´s Correction (multiple groups) was performed. For comparing stacked graphs (groups showing proportion of a whole) Fischer´s Exact test was used. $P$-value < 0.05 was considered to represent a statistically significant difference. $P$-values are reported in figures only for those comparisons from which biological insights were drawn, thus the absence of a $p$-value does not necessarily indicate that a difference is non-significant.

To support statements about the lack of differences between experimental groups (where $p$-values are >0.05) we computed the effect size of the differences using GPower (University of Düsseldorf) with these parameters: $t$-test tails = two, $\alpha$ = 0.05, power (1-$\beta$) = 0.8 and the sample size of both groups. Then, the smallest difference that would have been significant was calculated based on the effect size and the standard deviations and sample sizes of the experimental groups. These calculated smallest significant differences and the observed differences are reported in the Figure legends. If the calculated smallest significant difference was lower than differences reported for similar experiments in other studies, we concluded that our data sets had sufficient statistical power to detect similar effects. Thus, we consider that statements about the absence of differences are warranted in such cases.

## Reporting summary

Further information on research design is available in the Nature Portfolio Reporting Summary linked to this article.

# Data availability

scRNA seq data from cardiomyocytes is available under GEO accession number GSE130940. Microarray transcriptomics data is available as Supplementary Data File 1 and 2. Plasmids and transgenic lines will be shared on request. Source data are provided with this paper.

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

## Acknowledgements

We thank Doris Weber and Janet Köhler for their contributions to fish care, Guoxin Sun for help with qRT-PCRs, Laura Kellerer for rescuing the *bmp7a*−/− mutants and the core facility "Light microscopy" of the Medical faculty of Ulm University for support with imaging. We thank Prof. Steffen Just and Dr. Bernd Gahr for help with neonatal mouse cardiomyocyte cultures. We thank Prof. Didier Stainier for providing space and resources for the single-cell RNA-sequencing. The Weidinger lab acknowledges funding by the Deutsche Forschungsgemeinschaft (DFG, German Research Foundation) within the Collaborative Research Centre "Aging at Interfaces" (CRC1506; Project-ID 450627322), the CRC1149 "Danger Response, Disturbance Factors and Regenerative Potential After Acute Trauma" (Project-ID 251293561), and the CRC1279 "Exploiting the Human Peptidome for Novel Antimicrobial and Antic-ancer Agents" (Project-ID 316249678). M.D.V was also funded by an intramural grant of the Medical Faculty of Ulm University ("Baustein-Programm 3.2"). The research of K.S-K. is funded by the DFG within the Collaborative Research Centre (CRC1506; Project-ID 450627322) "Aging at Interfaces", the Collaborative Research Centre (CRC1149; Project-ID 251293561) "Danger Response, Disturbance Factors and Regenerative

Potential After Acute Trauma" and by the Graduate Training Group GRK 1789 "Cellular and Molecular Mechanisms in Ageing (CEMMA)". The D'Uva lab received funding from the European Union - NextGenerationEU through the Italian Ministry of University and Research under PNRR - M4C2-I1.3 Project PE_00000019 "HEAL ITALIA" to Gabriele Matteo D'Uva CUP J33C22002920006 and from the Italian Ministry of Health (RC-2024-2790614). L.W acknowledges funding by the DFG via project B3 in Research Training Group 2254 and project B03 in CRC 1506 (Project-ID 450627322). H.G acknowledges funding by the DFG via project B01 in CRC 1506 (Project-ID 450627322). D. P. P., H. F. M. M. I., J.N. are or were members of the International Graduate School in Molecular Medicine, Ulm University.

## Author contributions

Mohan Dalvoy Vasudevarao: Conceptualization (lead); Investigation—experiments using zebrafish, Formal analysis—zebrafish transcriptomics data; Visualization; Writing—review and editing. Denise Posadas Pena: Conceptualization (supporting); Investigation—experiments using zebrafish, Formal analysis—zebrafish transcriptomics data; Visualization; Writing—review and editing. Michaela Ihle: Investigation—DNA fiber spreading assays; Visualization; Writing—review and editing. Chiara Bongiovanni: Investigation—Mouse cardiomyocytes culture; Visualization; Writing—review and editing. Pallab Maity: Investigation—experiments in cultured fibroblasts; Visualization. Simone Redaelli: Methodology—creation and characterization of Smad6 expressing zebrafish transgenic lines; Writing—review and editing. Kathrin Happ: Methodology—creation and characterization of Smad6 expressing zebrafish transgenic lines. Dominik Geissler: Investigation—cardiomyocyte transcription data. Hossein Falah Mohammadi: Formal analysis—zebrafish transcriptomics data; Visualization. Melanie Rall-Scharpf: Resources—HSPCs. Julian Niemann: Investigation—HSPC CFU assay. Mathilda T.M. Mommersteeg: Conceptualization (supporting); Writing—review and editing. Chi-Chung Wu: Investigation—experiment using Noggin3 and Bmp2b overexpression transgenics; Writing—review and editing. Karin Scharffetter-Kochanek: Conceptualization (supporting); Funding Acquisition; Supervision. Gabriele D'Uva: Conceptualization (supporting); Funding Acquisition; Supervision; Writing—review and editing. Arica Beisaw: Resources—single cell data; Writing—review and editing. Mona Malek Mohammadi: Conceptualization (supporting); Funding Acquisition; Supervision. Hartmut Geiger: Conceptualization (supporting); Funding Acquisition; Supervision. Lisa Wiesmüller: Conceptualization (supporting); Funding Acquisition; Supervision; Writing – review and editing. Gilbert Weidinger: Conceptualization (lead); Funding Acquisition; Methodology—creation of Smad6 expressing zebrafish transgenic lines; Project Administration; Supervision; Visualization; Writing—original draft; Writing—review and editing.

## Funding

## Competing interests

The authors declare no competing interests.
