## [Peer Review file · Nature Communications]

BMP signaling promotes zebrafish heart regeneration via alleviation of replication stress

Corresponding Author: Professor Gilbert Weidinger

Version 0:

Reviewer comments:

Reviewer #1

(Remarks to the Author)

This is an interesting manuscript that has identified evidence of increased DNA damage in zebrafish cardiomyocytes after myocardial injury. Since zebrafish cardiomyocytes are actively synthesizing DNA during this time period, the authors suspected that the cause of the DNA damage was related to replication stress. The authors then determine that canonical BMP signaling impacts zebrafish cardiomyocyte DNA damage post-injury. They conclude that BMP signaling facilitates zebrafish myocardial regeneration through reduction of replication stress induced DNA damage.

The authors clearly show that cardiomyocyte DNA damage is induced post-injury. However, it is less clear if this DNA damage is secondary to replication stress since they did not assess for oxidative DNA damage. Likewise, despite many assays measuring cardiomyocyte DNA damage and cell cycle activity post-injury, there is minimal assessment of actual myocardial regeneration in the different experimental perturbations. The authors did show that dual inhibition of ATM and ATR reduces zebrafish myocardial regeneration. However, that result argues against the authors main conclusion that reduced DNA damage facilitates myocardial regeneration because ATM and ATR are activated by DNA damage. A low DNA damage state would lead to reduced ATM and ATR cellular activity, therefore mimicking the effect of ATM and ATR inhibition. Conversely, high levels of DNA damage lead to increased ATM and ATR activity. It will be important to reconcile how decreased ATM/ATR activity impairs zebrafish myocardial regeneration but reduced cardiomyocyte DNA damage from BMP signaling is reported by the authors to increase myocardial regeneration.

The other important component that needs better clarification is how BMP signaling can induce cardiomyocyte DNA synthesis but reduce replication stress induced DNA damage.

Fig 1 and 2:

DNA damage as assessed by gammaH2AX can occur secondary to both oxidative DNA damage and replication stress. The investigators are assuming that replication stress is occurring because DNA synthesis has also increased during the same time period. However, direct assessment of oxidative DNA damage using 8-OHdG should be performed to definitively rule out oxidative DNA damage as a contributor.

Fig 2C:

The investigators show that hydroxyurea can induce cardiomyocyte DNA damage which is thought to be secondary to replication stress Fig 2B and Fig S2. The primary conclusion of the manuscript is that increased replication stress impairs zebrafish myocardial regeneration. Therefore, the investigators should determine how hydroxyurea impacts myocardial regeneration after cryoinjury.

Fig S3A+B:

The investigators use rapamycin and starvation as a method to reduce cardiomyocyte DNA synthesis to help determine if increased DNA synthesis is causing the cardiomyocyte DNA damage. However, inhibiting MTOR signaling and starvation have broad effects on cellular physiology that are independent of DNA synthesis. Selective pharmacological inhibition of the

cardiomyocyte cell cycle using a CDK4/6 cell cycle inhibitor after cryoinjury would be a more direct approach at determining if reduced cardiomyocyte DNA synthesis leads to reduced cardiomyocyte DNA damage. A similar approach was performed in vivo in mammalian cardiomyocytes (J Am Heart Assoc. 2021 Aug 3;10(15):e021768).

Fig3b+3c and Fig S5A+B:

ATR is often selectively activated by replication stress in contrast to ATM (Nat Cell Biol. 2014 Jan; 16(1):2-9). Selective ATR activation has been previously shown to be induced in mammalian cardiomyocyte replication stress during hypertrophic cardiomyocyte growth (J Am Heart Assoc. 2021 Aug 3;10(15):e021768). Therefore, it is unclear why the investigators would simultaneously inhibit both ATR and ATM. Particularly since they show in Fig S5B that the ATM inhibitor synergizes with ionizing radiation (not replication stress). What impact did ATR inhibition alone have on wound healing, cardiomyocyte DNA synthesis and cardiomyocyte mitosis after cryoinjury? Are myocardial p53 levels altered by these perturbations?

Fig 4B:

The authors show that calcidiol increased cardiomyocyte mitosis (Fig 2E) after cryoinjury and it was associated with increased DNA damage (Fig 2F). Likewise, they show that MTOR and starvation reduce cardiomyocyte cell cycling and DNA damage (Fig S3A+B). The authors suggest that increased cardiomyocyte DNA synthesis after cryoinjury is inducing DNA damage secondary to replication stress. Previously, it was shown that increased BMP signaling induces zebrafish cardiomyocyte DNA cell cycle activity (hsp70l:bmp2b) and reduced BMP signaling (hsp70l:nog3) reduces cardiomyocyte DNA cell cycle activity and mitosis (Dev Cell 36, 36-49 (2016)). Therefore, how do the authors explain how inhibition of BMP signaling using hsp70l:nog3 was associated with increased cardiomyocyte DNA damage and overexpression of BMP signaling (hsp70l:bmp2b) caused less DNA damage in Fig 4B? If reduced BMP signaling leads to reduced cardiomyocyte cell cycle activity after injury as reported previously (Dev Cell 36, 36-49 (2016)), then there should be reduced replication stress induced DNA damage.

What effect did BMP inhibition with nog3 and activation with bmp2 have on post-injury cardiomyocyte DNA synthesis as assessed by Ki67 and mitosis?

Fig 5C:

The investigators have solely focused on cardiomyocyte DNA damage markers but what impact does this Smad6 overexpression model have on myocardial regeneration post-injury?

Fig 5D:

In Fig 5B and Fig S6A the investigators show that Smad6 overexpression almost completely abolishes p-Smad1/5/9 in cardiomyocytes. Therefore, how does Bmp2 overexpression still cause a large reduction of gH2AX+cardiomyocytes in the setting of Smad6 overexpression?

Fig 6B:

In an effort to understand how increased BMP signaling can induce cardiomyocyte cell activity and also reduce replication stress, the authors developed a theory that zebrafish cardiomyocytes 7d post-injury that are PCNA+ and Edu+ are in a cell cycle arrest state and zebrafish cardiomyocytes that are PCNA- and Edu+ have exited the cell cycle. Figure 2F shows high levels of PCNA+ cardiomyocytes at day 7 trending lower to day 14, so it is unclear how the authors have proven that PCNA+ cardiomyocytes are in a cell cycle arrested state? If the cardiomyocytes are PCNA+ then they are likely actively synthesizing DNA. Is there evidence that mitosis is reduced in the PCNA+ Edu+ cardiomyocyte population? Does Ki67+Edu+ provide similar results to PCNA+Edu+? If these cardiomyocytes are actively synthesizing DNA and not exiting the cell cycle, are they becoming polyploid?

Fig 6C+D:

The daily heat shocked WT gH2AX+Edu+ cardiomyocyte percentage is ~20% in Fig 6C, but the non-heat shocked WT gH2AX+Edu+ group is only ~5% in Fig 6D. This would suggest that heat shock alone has a dramatic impact on cardiomyocyte DNA damage post cyro injury. Heat shock is known to induce reactive oxygen species (Int J Hyperthermia. 2014 Nov;30(7):513-23). This would suggest that oxidative DNA damage may be a large contributor to the cardiomyocyte DNA damage detected. As mentioned earlier, this should be measured.

Fig 6E+F:

The lack of effect of Bmp2b overexpression on PCNA+Edu+ cardiomyocytes or EdU+gH2AX- cardiomyocytes in the absence of hydroxyurea argues against the authors primary conclusion that BMP signaling reduces cardiomyocyte replication stress. The authors suggest that endogenous BMP signaling is already sufficient to prevent replication stress. However, that would undermine the earlier results in Figure 4.

Fig 7A+C:

Does BMP7 or BMP2 induce DNA synthesis in the cardiomyocyte or fibroblasts?

Fig 8:

Does treatment of HSPCs or U2OS cells with BMP2 or BMP4 induce DNA synthesis or alter markers of DNA damage?

Reviewer #2

(Remarks to the Author)

The manuscript by Vasudevarao et al reports a role for BMP signaling in alleviation of replication stress in regenerating

zebrafish hearts. Gene expression studies confirmed by immunostaining provide evidence for DNA damage and repair responses during cardiac regeneration after injury in zebrafish. Gain and loss of function approaches demonstrate that BMP signaling can overcome the replication stress response and enable cardiac regeneration. Additional studies support the ability of BMP signaling to overcome drug-induced DNA damage in neonatal mouse cardiomyocytes and to promote DNA replication in human cell lines.

The zebrafish experiments are elegantly done and convincingly support the conclusion replication stress occurs during cardiac regeneration and can be overcome by BMP signaling using a variety of experimental approaches.

Comments

1. The conclusion as stated in the title has been demonstrated in zebrafish but not mice or other systems. The title should be revised to reflect this.
2. The studies in neonatal mouse cells show that BMP signaling can protect from HU-induced replication stress *in vitro*. However, there is no additional evidence presented to show that this mechanism contributes to neonatal heart regeneration *in vivo*. In addition, are alterations in this response related to loss of regenerative potential in mouse hearts in the weeks after birth?
3. The conclusion that replication stress does not affect CM dedifferentiation after injury is based on lack of expression of embryonic CM markers (Figure S4A), which is not definitive. Are cardiac sarcomeres broken down with HU treatment?
4. Similarly HU-treatments are needed to support conclusions regarding the role of replication stress on pPol2 and cellular senescence (Fig S4B, C).
5. The Discussion could be shorter and contains extensive speculation as well as material peripheral to the experimental findings.

Reviewer #3

(Remarks to the Author)

In this manuscript Vasudevarao and colleagues presented very interesting findings: the replication stress encountered by zebrafish cardiomyocytes (CMs) during heart regeneration and the important role of BMP signaling in mitigating this replication stress to facilitate cell cycle progression. Their study also illustrated that BMP signaling alleviated replication stress by accelerating DNA replication fork progression and by facilitating their re-start after replication stress-induced stalling in human cells subjected to hydroxyurea treatment. There are a few concerns need to be addressed before this manuscript is suitable for publication.

1. It is still confused, after reading the results and discussion, what are the differences between CM proliferation during regeneration and during normal development. And how does regeneration induce DNA damages in CMs? Can γ H2a.x staining represent all types of DNA damages, and more importantly, represent replication stress? The authors stated that replication stress is responsible for declining tissue renewal and repair in aged mammals, but the fraction of γ H2a.x+ CMs in regenerating hearts actually decreased in aged zebrafish. Is there any circumstance that cell experiencing replication stress but without γ H2a.x staining? The authors also stated that replication stress may be induced by the high demands on cell cycling, but later they suggested individual CMs do not undergo multiple rounds of cell division during regeneration. So what does the phrase "high CM cell cycle rates" (Page 9, line 43) mean?

2. In Figure 4B-F, the percentages of γ H2a.x+ CMs over border CMs in wild-type have huge variation, especially in Fig.4E the average is less than 4 which is much lower than the other panels and Fig.1E-G (around 10-20). The percentages in *bmp2b* OE, *bmp7b* OE and *bmp7a*^{-/-} are actually similar (around 5), the conclusion that the percentage of γ H2a.x+ CMs increases in *bmp7a*^{-/-} is merely due to the low percentage in wild-type in Fig.4E. This holds true for Figure 6C, D, the percentage increases in *bmp7a*^{-/-} is merely due to the low percentage in wild-type in Fig.6D.

3. The authors showed zebrafish CMs experience replication stress during regeneration but not physiological heart growth, and suggested that the ability to efficiently overcome replication stress is a key reason for the elevated capacity of adult zebrafish to regenerate the heart. When it comes to mouse experiments, people are eager to know if this holds true in neonatal mouse heart regeneration and if this may contribute to the differences in regenerative capability of neonatal and adult mouse hearts.

Other minor concerns:

1. In Figure 1A, the transcriptomic analysis is based on resection model while the rest of Figure 1 are cryoinjury model. The author could analyze RNA-seq dataset of cryoinjury from database and compare different time points.
2. Figure 2A, the author should also provide the percentage of γ H2a.x+ and γ H2a.x- in EdU+ CMs.
3. Figure 2C, the main text stated unperturbed regenerating hearts but in the diagram indicated HU treatment.
4. Do *bmp7a* and *bmp7b* have similar functions? The overexpression is using *bmp7b* while the knockout is *bmp7a*.
5. The title may be a little overstated, the current data only suggest BMP signaling promotes ZEBRAFISH heart regeneration via alleviation of replication stress.

Version 1:

Reviewer comments:

Reviewer #1

(Remarks to the Author)

The authors have performed multiple revisions that have significantly strengthened the manuscript. However, it would be helpful if they address one of the primary questions that was outlined in the initial review:

"The authors did show that dual inhibition of ATM and ATR reduces zebrafish myocardial regeneration. However, that result argues against the authors main conclusion that reduced DNA damage facilitates myocardial regeneration because ATM and ATR are activated by DNA damage. A low DNA damage state would lead to reduced ATM and ATR cellular activity, therefore mimicking the effect of ATM and ATR inhibition. Conversely, high levels of DNA damage lead to increased ATM and ATR activity. It will be important to reconcile how decreased ATM/ATR activity impairs zebrafish myocardial regeneration but reduced cardiomyocyte DNA damage from BMP signaling is reported by the authors to increase myocardial regeneration."

To reiterate, it is well established that ATR and ATM activity are increased by DNA damage. Therefore, increased cardiomyocyte replication stress DNA damage in the regenerating zebrafish heart should lead to increased ATR and/or ATM activity. If replication stress DNA damage is a negative regulator of zebrafish cardiomyocyte proliferation (as shown in the manuscript), then inhibition of the DNA damage induced kinases ATR and ATM would be predicted to increase zebrafish cardiomyocyte proliferation (mimicking a low DNA damage state). However, the authors data shows that reduction of ATR and ATM activity reduces zebrafish cardiomyocyte proliferation. How are these potentially contradictory results explained? It would be helpful to address this in the discussion section.

Reviewer #2

(Remarks to the Author)

The revised manuscript includes additional experiments that solidify the findings in regenerating zebrafish hearts. The manuscript and title have been edited to focus on BMP signaling and regenerative stress in zebrafish heart regeneration. These aspects of the manuscript are improved and address multiple reviewer comments.

The authors also attempted to extend their findings to neonatal mouse heart regeneration. I appreciate that these studies are challenging and that the results were inconclusive. A rigorous analysis of BMP signaling and replication stress in mammalian heart regeneration could easily be the focus of an entirely new manuscript that would be of high interest in a future study beyond the scope of the current manuscript.

Reviewer #3

(Remarks to the Author)

My concerns have been addressed.

Point-by-point response to the reviewers' comments

Reviewer #1

We thank reviewer 1 for their careful consideration of our work and for their thoughtful comments, which helped us to improve our manuscript and to strengthen our conclusions.

This is an interesting manuscript that has identified evidence of increased DNA damage in zebrafish cardiomyocytes after myocardial injury. Since zebrafish cardiomyocytes are actively synthesizing DNA during this time period, the authors suspected that the cause of the DNA damage was related to replication stress. The authors then determine that canonical BMP signaling impacts zebrafish cardiomyocyte DNA damage post-injury. They conclude that BMP signaling facilitates zebrafish myocardial regeneration through reduction of replication stress induced DNA damage.

The authors clearly show that cardiomyocyte DNA damage is induced post-injury. However, it is less clear if this DNA damage is secondary to replication stress since they did not assess for oxidative DNA damage. Likewise, despite many assays measuring cardiomyocyte DNA damage and cell cycle activity post-injury, there is minimal assessment of actual myocardial regeneration in the different experimental perturbations. The authors did show that dual inhibition of ATM and ATR reduces zebrafish myocardial regeneration. However, that result argues against the authors main conclusion that reduced DNA damage facilitates myocardial regeneration because ATM and ATR are activated by DNA damage. A low DNA damage state would lead to reduced ATM and ATR cellular activity, therefore mimicking the effect of ATM and ATR inhibition. Conversely, high levels of DNA damage lead to increased ATM and ATR activity. It will be important to reconcile how decreased ATM/ATR activity impairs zebrafish myocardial regeneration but reduced cardiomyocyte DNA damage from BMP signaling is reported by the authors to increase myocardial regeneration.

The other important component that needs better clarification is how BMP signaling can induce cardiomyocyte DNA synthesis but reduce replication stress induced DNA damage.

We have performed several experiments to address these issues, as detailed in the responses to the individual comments below.

Fig 1 and 2: DNA damage as assessed by gammaH2AX can occur secondary to both oxidative DNA damage and replication stress. The investigators are assuming that replication stress is occurring because DNA synthesis has also increased during the same time period. However, direct assessment of oxidative DNA damage using 8-OHdG should be performed to definitively rule out oxidative DNA damage as a contributor.

We have attempted immunofluorescence for 8-hydroxy-2-deoxyguanosine (8-OHdG) on cryosections of regenerating hearts, to address whether cardiomyocytes acquire oxidative DNA damage. However, despite several tweaks to the protocol and use of different antibodies we have not been able to obtain conclusive results. We have thus resorted to an ELISA for 8-OHdG, performed on DNA isolated from injured hearts to investigate whether there is an overall increase in oxidative DNA damage during regeneration (**new data in Figure 1H**). While positive controls where hydrogen peroxide was added to heart explant culture media for 1 hour showed a significant increase, there was no difference in 8-OHdG levels between uninjured and 7 dpi regenerating hearts.

We therefore conclude that the γ H2a.x accumulation that we observe specifically during regeneration is unlikely due to oxidative stress.

We describe these new data in lines 107-110 as follows:

“We assessed oxidative DNA damage using ELISA for 8-hydroxy-2'-deoxyguanosine (8-OHdG). Peroxide treatment of heart explants induced DNA oxidation but cryoinjury alone did not increase 8-OHdG levels compared to uninjured hearts (**Figure 1H**). These data suggest that oxidative DNA damage is an unlikely cause of γ H2a.x positivity in CMs.”.

Fig 2C: The investigators show that hydroxyurea can induce cardiomyocyte DNA damage which is thought to be secondary to replication stress Fig 2B and Fig S2. The primary conclusion of the manuscript is that increased replication stress impairs zebrafish myocardial regeneration. Therefore, the investigators should determine how hydroxyurea impacts myocardial regeneration after cryoinjury.

We concur with the reviewer's view that long term assays with induced replicative stress should be tried to check the effect on regeneration. However, our attempts to perform such experiments have unfortunately resulted in severe health issues of the treated fish within a week of treatment, in particular anemia, presumably due to suppression of hematopoietic stem cell proliferation. Thus, for ethical reasons we had to terminate the experiments at a time-point that is too early to assess myocardial regeneration. Use of lower doses of HU, which might be better tolerated, did not induce CM replication stress as measured by γ H2a.x accumulation. Thus, in the absence of (genetic) tools that would allow us to induce replication stress specifically in CMs, we can unfortunately not test the consequences of increased replication stress on morphological heart regeneration / scarring, which can only be read out from 21 dpi onwards.

Fig S3A+B: The investigators use rapamycin and starvation as a method to reduce cardiomyocyte DNA synthesis to help determine if increased DNA synthesis is causing the cardiomyocyte DNA damage. However, inhibiting MTOR signaling and starvation have broad effects on cellular physiology that are independent of DNA synthesis. Selective pharmacological inhibition of the cardiomyocyte cell cycle using a CDK4/6 cell cycle inhibitor after cryoinjury would be a more direct approach at determining if reduced cardiomyocyte DNA synthesis leads to reduced cardiomyocyte DNA damage. A similar approach was performed in vivo in mammalian cardiomyocytes (J Am Heart Assoc. 2021 Aug 3;10(15):e021768).

We thank the reviewer for this excellent suggestion. We have treated cryoinjured fish for 24 hours with the CDK4/6 inhibitor PD-0332991, and have analyzed CM mitosis and replication stress at 7 dpi (**new data in Figure 2E and F**). We observed that the CDK inhibitor decreased the fraction of pH3+ mitotic CMs (**Figure 2E**), and at the same time also reduced the fraction of γ H2a.x+ CMs (**Figure 2F**). These results strongly support our model that γ H2a.x+ is indicative of replication stress. We describe these results in lines 147-150 as follows:

“To further support the correlation between CM cycling and γ H2a.x accumulation in regenerating hearts, we directly inhibited cell cycle progression using the CDK4/6 inhibitor PD-0332991. One day treatment was sufficient to strongly decrease the number of mitotic pH3+ CMs and the number of γ H2a.x+ CMs at 7 dpi (**Figure 2E, F**)”.

Fig3b+3c and Fig S5A+B: ATR is often selectively activated by replication stress in contrast to ATM (Nat Cell Biol. 2014 Jan; 16(1):2-9). Selective ATR activation has been previously shown to be induced in mammalian cardiomyocyte replication stress during hypertrophic cardiomyocyte growth (J Am Heart Assoc. 2021 Aug 3;10(15):e021768). Therefore, it is unclear why the investigators would simultaneously inhibit both ATR and ATM. Particularly since they show in Fig S5B that the ATM inhibitor synergizes with ionizing radiation (not replication stress). What impact did ATR inhibition alone have on wound healing, cardiomyocyte DNA synthesis and cardiomyocyte mitosis after cryoinjury? Are myocardial p53 levels altered by these perturbations?

Following the reviewer’s suggestion, we have evaluated the impact of ATR inhibition on CM mitosis at 7 dpi after a 24-hour treatment. This was indeed sufficient to interfere with CM mitosis (**new data in Figure 3B**), supporting our hypothesis that alleviation of replication stress is required for regenerative CM proliferation. Interestingly, the decrease of pH3+ CMs observed by treatment with ATRi was less (38%) than the one caused by combined treatment with ATM and ATR inhibitors (54%, **Figure 3C**). This is in agreement with findings that ATM and ATR kinase activated DNA repair pathways can crosstalk. For example, it has been shown that ATM kinase contributes to specific replicative stress response in an ATR-dependent manner (doi.org/10.1038/sj.emboj.7601446). Therefore, we used dual ATM and ATR inhibition for studying the long-term effects of suppression of DNA damage response pathways on morphological heart regeneration and scarring (**Figure 3D**). Regarding p53 levels: Due to technical issues, specifically the lack of reliably functioning antibodies for zebrafish (we have attempted both immunostainings as well as immunoblots using two different antibodies), despite our best efforts, we could not evaluate the levels of p53 in our experiments. While we acknowledge its importance, we are unable to address this question at this time.

We describe these new data in lines 203-204 as follows:

“Intriguingly, we found that treatment of fish with the ATR inhibitor VE821 for 24 h was sufficient to reduce CM mitosis in regenerating hearts at 7 dpi (**Figure 3B**)”.

Fig 4B: Q: The authors show that calcidiol increased cardiomyocyte mitosis (Fig 2E) after cryoinjury and it was associated with increased DNA damage (Fig 2F). Likewise, they show that MTOR and starvation reduce cardiomyocyte cell cycling and DNA damage (Fig S3A+B). The authors suggest that increased cardiomyocyte DNA synthesis after cryoinjury is inducing DNA damage secondary to replication stress. Previously, it was shown that increased BMP signaling induces zebrafish cardiomyocyte DNA cell cycle activity (hsp70l:bmp2b) and reduced BMP signaling (hsp70l:nog3) reduces cardiomyocyte DNA cell cycle activity and mitosis (Dev Cell 36, 36-49 (2016)). Therefore, how do the authors explain how inhibition of BMP signaling using hsp70l:nog3 was associated with increased cardiomyocyte DNA damage and overexpression of BMP signaling (hsp70l:bmp2b) caused less DNA damage in Fig 4B? If reduced BMP signaling leads to reduced cardiomyocyte cell cycle

*activity after injury as reported previously (Dev Cell 36, 36-49 (2016)), then there should be reduced replication stress induced DNA damage. What effect did BMP inhibition with *nog3* and activation with *bmp2* have on post-injury cardiomyocyte DNA synthesis as assessed by Ki67 and mitosis?*

It is indeed striking that BMP signaling behaves very differently than all other pro-regenerative pathways that we have tested. As pointed out by the reviewer, in all other interventions where we either experimentally increase or decrease CM cell cycling, we observe increased or decreased γ H2a.x accumulation as well, strengthening our conclusion that CMs experience replication stress. We also have preliminary evidence that the overactivation of other pro-regenerative pathways beyond Vitamin D, which are known to enhance CM cycling, also increases γ H2a.x accumulation (we tested Wnt/beta-catenin and Nf-kappaB signaling). Thus, it is all the more interesting that BMP signaling has the opposite effect. We have indeed found and published that increased BMP signaling induces zebrafish cardiomyocyte DNA cell cycle activity (via *Bmp2* overexpression) and reduced BMP signaling (using *Noggin3* overexpression) reduces cardiomyocyte DNA cell cycle activity and mitosis (Wu *et al.* 2016, *Dev Cell* 36, 36-49). More recently, we also showed in collaboration with the D'UVa lab that overexpression of *bmp7* is sufficient to increase CM cycling, while *bmp7a* mutants display decreased CM cycling activity (Bongiovanni *et al.*, *Cell Rep.* 2024, doi: 10.1016/j.celrep.2024.114162). Yet, in this manuscript, we provide strong evidence that BMP gain-of-function (via overexpression of *Bmp2* or *Bmp7*) decreases replication stress, while BMP loss-of-function (via overexpression of *Noggin3*, *Smad6* or in *bmp7a* mutants) increases replication stress.

We interpret these intriguing findings to indicate that BMP signaling has a dual function: it promotes CM cell cycling (likely via allowing more CMs at the wound border to enter the cell cycle), but at the same time strongly decreases replication stress. Specifically, we provide evidence that BMP signaling promotes stress-free cell cycling activity (previously existing data in Figure 6E and F). Of note, we show that the anti-replicative stress function of BMP signaling is conserved in mouse CMs and a variety of human cells (we have added **new data showing this also for human primary hematopoietic stem and progenitor cells, Figure 7E**). Interestingly, in human cells we find cases where BMP signaling ONLY reduces replication stress without increasing the overall rate of cell cycling. Our **new data in Supplementary Figure 9** show that the treatment with BMP ligands does not increase EdU incorporation of human primary fibroblasts, while it strongly protects them from HU-induced replication stress (previously existing data, now in **Figure 7D**). Overall, we think that the anti-replication stress role of BMP signaling that we describe here is not add odds with previous findings, but rather point to an interesting additional function of the pathway.

*Fig 5C: The investigators have solely focused on cardiomyocyte DNA damage markers but what impact does this *Smad6* overexpression model have on myocardial regeneration post-injury?*

We have performed an additional experiment where we performed once daily heat-shocks of wild-types and sibling *hsp70l:nt-p2a-smad6b* fish for 21 days, which should result in long-term but intermittent BMP-SMAD pathway inhibition (more than one daily heat-shock is unfortunately not well tolerated). We scored for wound size and collagen content (**new data in Supplementary Figure 8B and Figure 5E**). Wound size is the same in both groups in hearts harvested at 3 dpi, showing that the wounding conditions were similar. Yet, at 21 dpi *Smad6* overexpressing fish had moderately, but significantly bigger wounds (**Supplementary Figure 8B**). Importantly however, these wounds contained strikingly more collagen (determined as the blue areas in AFOG staining) than the ones in wild-types (**Figure 5E**). We conclude that BMP/*Smad* signaling is required for morphological heart

regeneration, in particular for scar resorption, which is in agreement with our previously published findings using long-term overexpression of Noggin3 (*Wu et al. 2016, Dev Cell 36, 36-49*). We describe these new data in lines 263-265 as follows:

“While wound sizes did not differ at 3 dpi, *hsp70l:nT-p2a-smad6b* cryoinjured hearts displayed larger wounds at 21 dpi (**Supplementary Figure 8B**). Importantly, these wounds contained considerably more collagen (**Figure 5E**)”.

Fig 5D: Q: In Fig 5B and Fig S6A the investigators show that Smad6 overexpression almost completely abolishes p-Smad1/5/9 in cardiomyocytes. Therefore, how does Bmp2 overexpression still cause a large reduction of γ H2AX+cardiomyocytes in the setting of Smad6 overexpression?

Of note, while we show in Fig. 5B that p-Smad1/5/9 is very efficiently suppressed in the CMs that express the *hsp70l:nT-p2a-smad6b* transgene (marked by nuclear Tomato, nT+), expression is mosaic, such that a considerable fraction of CMs show normal pSmad1/5/9 levels (nT – CMs in Fig. 5B). While *smad6b* expression thus clearly acts cell-autonomously, and unfortunately only in a subset of CMs, we cannot tell which CMs are affected by *bmp2b* overexpression in *hsp70l:bmp2b* transgenics. First, the transgene does not contain a marker for detection of the transgene-expressing cells, and secondly is the BMP ligand expected to act non-cell-autonomously anyways. Therefore, when we combine both transgenes for the rescue experiments, we cannot expect the Smad6 overexpression to be fully blocking the effects of BMP overexpression, since a considerable fraction of CMs will not express Smad6; yet these CMs could express the BMP transgene themselves or might see BMP ligand expressed by other CMs. Despite this caveat, we nevertheless think that this experiment is a good indication for the fact that BMP ligand overexpression acts, at least largely, through Smad signaling, since the effect of BMP signaling in decreasing γ H2a.x is completely reversed in the double transgenics.

Fig 6B: In an effort to understand how increased BMP signaling can induce cardiomyocyte cell activity and also reduce replication stress, the authors developed a theory that zebrafish cardiomyocytes 7d post-injury that are PCNA+ and Edu+ are in a cell cycle arrest state and zebrafish cardiomyocytes that are PCNA- and Edu+ have exited the cell cycle. Figure 2F shows high levels of PCNA+ cardiomyocytes at day 7 trending lower to the day 14, so it is unclear how the authors have proven that PCNA+ cardiomyocytes are in a cell cycle arrested state? If the cardiomyocytes are PCNA+ then they are likely actively synthesizing DNA. Is there evidence that mitosis is reduced in the PCNA+ Edu+ cardiomyocyte population? Does Ki67+Edu+ provide similar results to PCNA+Edu+? If these cardiomyocytes are actively synthesizing DNA and not exiting the cell cycle, are they becoming polyploid?

We appreciate the reviewer’s concerns and we agree that it is an oversimplification to apply the word “arrest” to all cells which are EdU+ PCNA+ at 7 dpi. We have rewritten the paragraph and now distinguish CMs that have incorporated EdU, but are PCNA negative at 7 dpi, as those that have “exited” the cell cycle, while we describe those that have incorporated EdU and are still PCNA+ at 7 dpi as those with “ongoing” cell cycle. These will include CMs also seen in wild-types that continue to cycle, but also CMs where cell cycle completion is delayed due to replication stress.

Unfortunately, we are unable to perform Ki67 staining due to the lack of functional antibodies currently available for zebrafish. The question regarding ploidy is an interesting one. It is clear that there are no altered ploidy levels in CMs during unperturbed zebrafish heart regeneration (10.1016/j.devcel.2018.01.021). However, whether loss of BMP signaling results in enhanced ploidy states has not been checked and is something we would like to investigate further. Unfortunately, this requires isolation of cardiomyocytes by FACS or density gradient centrifugation, both of which have not been working well enough for us so far.

We have changed the results text to avoid the term “arrested” cell cycle at several locations including this piece of text from line 278-284:

“Our previous quantification and modeling of CM number increase during regeneration has suggested that individual CMs do not undergo multiple rounds of cell division {Bertozzi, 2021 #161}; thus, a minority of CMs that are cycling at 4, 5 or 6 dpi are expected to still be cycling at 7 dpi, rather many will have withdrawn from the cell cycle by that time. Building on this concept, we define EdU+ PCNA+ CMs at 7 dpi as those with ongoing cell cycle, which are possibly delayed in exiting the cycle, while CMs that are only EdU+ as those that have completed cycling (**Figure 6B**)”

Fig 6C+D: The daily heat shocked WT γ H2AX+EdU+ cardiomyocyte percentage is ~20% in Fig 6C, but the non-heat shocked WT γ H2AX+EdU+ group is only ~5% in Fig 6D. This would suggest that heat shock alone has a dramatic impact on cardiomyocyte DNA damage post cryoinjury. Heat shock is known to induce reactive oxygen species (Int J Hyperthermia. 2014 Nov;30(7):513-23). This would suggest that oxidative DNA damage may be a large contributor to the cardiomyocyte DNA damage detected. As mentioned earlier, this should be measured.

Variation in the fraction of γ H2a.x+ CMs between experiments is indeed relatively high. We have tested the reviewer’s suggestion that heat-shock is to blame (see **Figure A in this response to reviewer’s**). We compared γ H2a.x levels in CMs of heat shocked wild type fish with non-heat shocked wild-type fish, and fortunately did not observe a difference. We suspect that the experiment-to-experiment variation of γ H2a.x levels reflect multiple variables. First, we have noted that fresh batches of antibodies produce better signal-to-noise during immunostainings, which allow us to count more CMs as being clearly positive. Since we cannot order fresh antibody for every experiment, some variations are due to stainings with different age of antibody. Secondly, we have shown that γ H2a.x levels correlate with the rate of CM cycling (except under conditions where we manipulate BMP signaling). We and others have repeatedly observed that the fraction of CMs entering the cell cycle at the wound border can vary substantially between groups of fish, despite seemingly identical injury conditions. One major factor explaining these variations appears to be the genetic background. Our collaborator Mathilda Mommersteeg has kindly provided unpublished data showing that different wild type strains respond differently with respect to PCNA levels upon injury (**Figure B in this response to reviewer’s**). Since we work with several transgenic lines, which were provided to us in different genetic backgrounds, our experiments were performed in fish of varying backgrounds.

Figure A. Heat-shocks applied to wild-type fish once daily do not affect γ H2a.x accumulation in CMs at 7 dpi.

[keda ted]

Figure B. The fraction of cycling CMs (detected by PCNA immunofluorescence) varies considerably between different wild-type strains at 7 and 21 dpi after cryoinjury.

While we thus cannot avoid differences in CM cycling and replication stress between experiments, we take extreme care to make sure that our controls and experimental fish WITHIN each experiment are as similar as possible. We only use controls (e.g., wild-types) that are siblings of the experimental fish (and thus will share the same genetic background) and raise the fish together, such that any environmental influences apply to all of them. We genotype fish only very shortly prior to the experiment or perform the experiments on unidentified fish to make sure that their feeding status, stocking density etc. is again as similar as possible. In addition, we take care to keep control and experimental fish at defined and identical conditions after cryoinjury (during regeneration), in particular always at identical stocking density. Finally, samples for control and experimental fish are always processed at the same time with identical reagents, e.g. antibody batches, and are imaged at the same time with identical imaging parameters.

Thus, despite the issue of experiment-to-experiment variations, we are confident that the results that we report based on comparison of fish within experiments are sound.

Fig 6E+F: The lack of effect of Bmp2b overexpression on PCNA+Edu+ cardiomyocytes or Edu+gH2AX-cardiomyocytes in the absence of hydroxyurea argues against the authors primary conclusion that BMP signaling reduces cardiomyocyte replication stress. The authors suggest that endogenous BMP signaling is already sufficient to prevent replication stress. However, that would undermine the earlier results in Figure 4.

We thank the reviewer for raising this point; we realized that we had made a mistake in reporting p-values in Fig. 6E and 6F. In Fig. 6F, the fraction of “stress-free” replicating Edu+ γ H2a.x- CMS is clearly different between the wild-type and BMP overexpressing, untreated fish, as seen by non-

overlapping 95% confidence interval error bars, yet p-value was reported as 0.2 in the previous version of the manuscript. We have gone back and re-calculated p-values and find that this difference is indeed very significant (**updated Fig. 6F**). Thus, BMP gain-of-function is sufficient to promote “stress-free” CM replication also in the absence of hydroxyurea. We have also updated p-values in Fig. 6E, yet there is still no significant difference in the population of CMs with “ongoing” cell cycle (PCNA+EdU+) between wild-type and BMP2b overexpressing untreated fish. We assume that there are two possible explanations why we can detect an effect of BMP gain-of-function in the absence of hydroxyurea only on the “stress-free” CM population, but not the “ongoing cell cycle” population. First, the latter assay is less sensitive since the underlying data are noisier (compare Fig. 6E and 6F). Secondly, the PCNA+EdU+ population of CMs is likely a mixture of cells that undergo unperturbed cell cycling and those that are delayed due to replication stress, as discussed above, which also limits the sensitivity of detecting differences in the latter subpopulation. We have removed the statement that endogenous BMP signaling might be sufficient to reduce replication stress and describe these results from line 300-305 as follows:

“While *bmp2b* overexpression was not sufficient to decrease the fraction of PCNA+ EdU+ CMs in otherwise unperturbed conditions, it alleviated the increase in their numbers caused by HU treatment (**Figure 6E**). Intriguingly, *bmp2b* overexpression was able to increase the fraction of EdU+ γ H2a.x- CMs that experience “stress-free” replication both under unperturbed conditions and upon additional replication stress induced by HU (**Figure 6F**)”.

Fig 7A+C: Does BMP7 or BMP2 induce DNA synthesis in the cardiomyocyte or fibroblasts?

We have previously published that in cultured mouse neonatal ventricular CMs, BMP7 does induce DNA synthesis (DOI: 10.1016/j.celrep.2024.114162). As for fibroblasts, we have now included data that a combinatorial treatment (BMP2+BMP4) did not alter EdU incorporation (**new data in Supplementary Figure 9**). The ability of BMPs to stimulate cell cycling appears to vary between cell types. The strong effect of BMP signaling gain-of-function on suppression of hydroxyurea-induced γ H2a.x in the fibroblasts in the absence of an effect on cycling rates strengthens our conclusion that it acts to alleviate or protect cells from replication stress. We describe these new data in lines 321-323 as follows:

“Likewise, BMP2 and BMP4 treatment suppressed the ability of HU to induce γ H2a.x in neonatal human foreskin dermal fibroblasts (**Figure 7D**), while it did not enhance replication as observed by EdU incorporation (**Supplementary Figure 9A**)”.

Fig 8: Does treatment of HSPCs or U2OS cells with BMP2 or BMP4 induce DNA synthesis or alter markers of DNA damage?

We have added **new data in Figure 7C**, where we quantified pan-nuclear γ H2a.x in U2OS cells, either in the presence of HU alone or combined with BMP ligand pre-treatment. BMP ligands pre-treatment lead to a drastic reduction of HU induced pan-nuclear γ H2a.x. Thus, we could observe that BMP signaling reduces replication stress also in U2OS cells. Overall, we see this effect in zebrafish

CMs, mouse CMs, human primary fibroblasts, human primary HSPCs (**new data in Figure 7E**), and U2OS cells. We describe these new data in lines 319-321 as follows:

“In the human U2OS cell line, combined treatment with recombinant BMP2 and BMP4 proteins was sufficient to completely abrogate the induction of γ H2a.x in response to HU treatment (**Figure 7C**)”.

Reviewer #2:

We thank reviewer 2 for their careful consideration of our work and for their thoughtful comments, which helped us to improve our manuscript and to strengthen our conclusions

The manuscript by Vasudevarao et al reports a role for BMP signaling in alleviation of replication stress in regenerating zebrafish hearts. Gene expression studies confirmed by immunostaining provide evidence for DNA damage and repair responses during cardiac regeneration after injury in zebrafish. Gain and loss of function approaches demonstrate that BMP signaling can overcome the replication stress response and enable cardiac regeneration. Additional studies support the ability of BMP signaling to overcome drug-induced DNA damage in neonatal mouse cardiomyocytes and to promote DNA replication in human cell lines.

The zebrafish experiments are elegantly done and convincingly support the conclusion replication stress occurs during cardiac regeneration and can be overcome by BMP signaling using a variety of experimental approaches.

We thank the reviewer for their positive overall assessment of our work.

1. The conclusion as stated in the title has been demonstrated in zebrafish but not mice or other systems. The title should be revised to reflect this.

We have revised the title to reflect the above concern. It now reads “BMP signaling promotes zebrafish heart regeneration via alleviation of replication stress”.

The studies in neonatal mouse cells show that BMP signaling can protect from HU-induced replication stress in vitro. However, there is no additional evidence presented to show that this mechanism contributes to neonatal heart regeneration in vivo. In addition, are alterations in this response related to loss of regenerative potential in mouse hearts in the weeks after birth?

We agree with the reviewer that it would be very interesting to know to which extent replication stress occurs during neonatal mouse heart regeneration and whether BMP signaling plays a role in alleviating it. Unfortunately, the experiments we have performed to answer these questions have been inconclusive. In collaboration with Mona Malek Mohammadi (University of Bonn) we have first tried to assess by immunofluorescence against γ H2a.x whether neonatal mouse cardiomyocytes display signs of DNA damage/replication stress. Unfortunately, we were only able to obtain non-specific staining in all CM nuclei. We thus resorted to Western blots for γ H2a.x. To avoid the issue that wounded/infarcted tissue might contain other cell types with damaged DNA, we concentrated our analysis on the non-injured right ventricle, which displays robust upregulation of CM cell cycling

in mice where the left ventricle was infarcted (see **Figure C** in this response to reviewer's from Hu et al., JCI Insight. 2024;9(5):e176281). doi: 10.1172/jci.insight.176281). Since the levels of additional CM cycling in the right ventricle induced by infarction of the left ventricle are actually similar to the levels in the left ventricle, if not higher, we assume that replication stress should be detectable there as well, if it poses a challenge to CM cycling in the neonatal mouse heart. We performed Western blots for γ H2a.x and the nuclear marker H3 on right ventricles of mice that underwent sham injury or myocardial infarction and on groups that were treated with the BMP signaling inhibitor LDN193189 (**Figure D in this response to reviewer's**).

As a positive control for our ability to detect DNA damage with this antibody in mouse cells, we also blotted irradiated mouse B-lymphocyte (CLL) cells (**Figure D in this response to reviewer's**). While the positive controls worked, we had a hard time convincing ourselves that the weak and fuzzy bands detected in any of the mouse hearts truly reflect γ H2a.x levels (**Figure E in this response to reviewer's**).

When we nevertheless tried to amplify the signals, we could not detect differences between the four groups. We also performed western blots for the replication stress marker p-Rpa32 and were unable to detect any bands. Unfortunately, we are not able to perform an *in vivo* positive-control for γ H2a.x and p-Rpa32 accumulation like hydroxyurea treatment of mice due to the lack of a corresponding animal license. Thus, these results remain inconclusive. We hope that the reviewer agrees that clarifying whether CM replication stress exists in regenerating neonatal mouse hearts is beyond the scope of this manuscript, since it requires additional animal licenses that are very time-consuming to obtain in Germany.

We attempted to gain evidence for or against the existence of replication stress by checking whether upregulation of DNA damage related genes during neonatal mouse heart

~~[keda ted]~~

Figure C. Robust induction of CM cycling in the right ventricle upon myocardial infarction of the left ventricle of neonatal mice. From Hu et al., JCI Insight. 2024;9(5):e176281

Figure D. Western blots for γ H2a.x and nuclear H3 using non-irradiated and irradiated mouse B-lymphocyte CLL line.

Figure E. Western blots for γ H2a.x and the nuclear marker H3 on right ventricles of neonatal mouse (P1) after sham injury (sham) or myocardial infarction (MI) of the left ventricle. They were either injected with water (solvent) or with the BMP signaling inhibitor LDN193189.

regeneration can be observed on an existing dataset (GSE123863) from sham injury vs infarction (Wang Z, et al., 2019 <https://doi.org/10.1073/pnas.1905824116>). Upon analyzing gene ontology for differentially expressed genes we could not detect a significant enrichment for DNA damage. Curiously, as in zebrafish we could not find enrichment for DNA replication either. Thus, since we cannot clearly show whether replication stress exists in the regenerative stage of injured neonatal mouse hearts, we can also not answer whether any changes in DNA damage response pathway activities are associated with the loss of regenerative potential.

However, it is worth noting that we have previously published that the levels of BMP7 decrease with neonatal heart growth in mouse (Bongiovanni et al. 2024, Cell Rep, doi: 10.1016/j.celrep.2024.114162). BMP7 stimulates CM proliferation *in-vitro* and *in-vivo* after mouse neonatal heart injury and adult myocardial infarction and its downregulation is thus clearly correlated with the loss of regenerative potential shortly after birth. In this manuscript we show that Bmp7 is an essential regulator of replication stress in the zebrafish heart as well, and that the role of BMP signaling in alleviation of replication stress is well conserved between zebrafish and mammals. Thus, it is tempting to speculate that BMP7 promotes neonatal mouse heart regeneration at least in part also because it does alleviate replication stress.

The conclusion that replication stress does not affect CM dedifferentiation after injury is based on lack of expression of embryonic CM markers (Figure S4A), which is not definitive. Are cardiac sarcomeres broken down with HU treatment?

We have now performed an additional experiment where CM sarcomeric disassembly was evaluated using alpha-actinin staining in the presence or absence of HU (**new data in Supplementary Figure 4B**). The data confirms our previous conclusion that HU treatment is not sufficient to alter CM dedifferentiation. We describe these new results starting at line 160-164 as follows:

“Induction of exogenous replication stress via HU treatment did not affect three readouts of CM dedifferentiation, namely the upregulation of a transgene reporting the activity of regulatory regions of the CM progenitor marker *gata4*, the upregulation of an embryonic myosin (embMHC) or the disassembly of sarcomeres (**Supplementary Figure 4A, B**). This indicates that CM dedifferentiation occurs independently and likely upstream of CM cell cycling, which is reduced by HU”.

Similarly HU-treatments are needed to support conclusions regarding the role of replication stress on pPol2 and cellular senescence (Fig S4B, C).

Our measurements of pPol2 were part of our effort to identify the endogenous molecular reasons for the replication stress experienced by wound border CMs. Within the limits of sensitivity of these assays we exclude that conflicts between transcription and replication forks represent a likely reason, since we cannot detect any global upregulation of transcription in wound border CMs. We are not sure how hydroxyurea treatment would help to support or disprove this conclusion. Hydroxyurea is a well-established replication stress inducer; however, it has not been reported to affect global transcription. Given that HU itself is unlikely to influence global transcription, we

believe that measuring p-Pol2 under HU treatment would not provide a meaningful insight. We feel this experiment would not add to the data and have currently chosen to omit it during revision. To address the role of HU on senescence we have added new data (**Supplementary Figure 5C**) where we have treated fish with HU for 7 days and performed beta-galactosidase staining. We do not observe any senescence in the spared myocardial area. We conclude that prolonged replication stress alone is not sufficient to drive CMs into senescence. We describe these new results starting at line 186-188 as follows:

“Interestingly, induction of exogenous replication stress by HU treatment for 7 days was not sufficient to induce CM senescence by 7 dpi (**Supplementary Figure 5C**)”.

The Discussion could be shorter and contains extensive speculation as well as material peripheral to the experimental findings.

We thank the reviewer for the suggestion. We have significantly reduced the length of this part of the manuscript.

Reviewer #3:

We thank reviewer 3 for their careful consideration of our work and for their thoughtful comments, which helped us to improve our manuscript and to strengthen our conclusions.

In this manuscript Vasudevarao and colleagues presented very interesting findings: the replication stress encountered by zebrafish cardiomyocytes (CMs) during heart regeneration and the important role of BMP signaling in mitigating this replication stress to facilitate cell cycle progression. Their study also illustrated that BMP signaling alleviated replication stress by accelerating DNA replication fork progression and by facilitating their re-start after replication stress-induced stalling in human cells subjected to hydroxyurea treatment. There are a few concerns need to be addressed before this manuscript is suitable for publication.

We thank the reviewer for their positive assessment of our work, and have addressed the concerns below.

It is still confused, after reading the results and discussion, what are the differences between CM proliferation during regeneration and during normal development. And how does regeneration induce DNA damages in CMs? Can γ H2a.x staining represent all types of DNA damages, and more importantly, represent replication stress? The authors stated that replication stress is responsible for declining tissue renewal and repair in aged mammals, but the fraction of γ H2a.x+ CMs in regenerating hearts actually decreased in aged zebrafish. Is there any circumstance that cell experiencing replication stress but without γ H2a.x staining? The authors also stated that replication stress may be induced by the high demands on cell cycling, but later they suggested individual CMs do not undergo multiple rounds of cell division during regeneration. So what does the phrase “high CM cell cycle rates” (Page 9, line 43) mean?

Our results clearly show that replication stress is restricted to cycling cardiomyocytes during regeneration, but does not occur during normal development or physiological heart growth in adults. Cycling rates (that is the fraction of CMs that are in the cell cycle) are typically higher during regeneration than during normal and physiological growth. In adults, CMs usually do not cycle, but heart growth can be stimulated by providing highly favorable growth conditions, mainly by sudden decrease in fish density. While CM cycling can be detected under these conditions, the rates are much lower than those observed during heart regeneration. The phrase “high demands on cell cycling” was meant to describe the high number of CMs that need to cycle for regeneration to be possible. In the case of adults, this entails recruitment of previously quiescent CMs to the cell cycle. We do not think that replication stress during normal development or physiological growth goes undetected. γ H2a.x accumulates in response to many forms of DNA damage, and not only upon replication stress, but we are not aware of published cases where replication stress occurs in the absence of γ H2a.x accumulation. Depending on whether γ H2a.x localization is pan-nuclear or as foci, it is possible to discern replication stress from DNA double strand breaks since in replication stress, there is diffuse staining all through the nucleus (at least in certain cell types). We proved that it is possible to detect replication stress in CMs by using HU treatment to elicit pan-nuclear staining in proliferating CMs (**Supplementary Fig. 2E**), and found that the vast majority of CMs that

accumulate γ H2a.x during regeneration do so in a pan-nuclear manner (**Figure 2C**). Also, we detect replication stress via the more specific marker p-Rpa32 in regenerating hearts as well (**Figure 2D**). We agree with the reviewer that the molecular reason for why CMs only experience replication stress during regeneration remains unclear; despite several efforts we have not been able to identify it. Of note, also in salamander limbs, where replication stress has been reported during regeneration as well, the underlying molecular cause is unknown. One possible explanation for the appearance of replication stress specifically only during regeneration is that the high fraction of cycling CMs puts limits on the replication machinery, including the availability of nucleotides. Unfortunately, we are not aware of how to test this in vivo. We have tested the hypothesis that conflicts of replication forks with transcription might occur during regeneration, but could not detect any global increases in transcription in wound border CMs during regeneration, which makes this explanation unlikely.

With respect to age-related γ H2a.x, we wish to highlight that irrespective of the age of the fish, heart injury consistently induces replication stress in CMs. However, differences in absolute levels between experiments (time points in **Figure 2H**) could be due to assay-related issues like the batch of antibody and the quality of staining, since these samples could of course not be processed at the same time (see the response to the next concern below). While we can very well control for such factors within an experiment, such that we are very confident in the differences that we report between uninjured and regenerating hearts at each time-point in **Figure 2H**, we are not prepared to attach too much meaning to the seemingly declining fraction of γ H2a.x+ CMs between 6 months and 2-year-old fish. We have modified **Figure 2H** and have removed the p-value between 6 months and 2-year old injured hearts to reflect this and we do not conclude that there is less replication stress in older regenerating hearts. However, the fact that aging decreases regenerative capacity of the zebrafish heart is published elsewhere (Reuter et al, Cells 2022. doi: 10.3390/cells11030345). Since we generally observe that CM cycling rates strongly correlate with the rate of γ H2a.x+ CMs (which supports our hypothesis that γ H2a.x accumulates due to replication stress), it is possible that the fraction of γ H2a.x+ CMs does indeed decrease in older fish that more poorly regenerate and thus display lower CM cycling rates. At any rate, the relationship between aging, replication stress and regeneration remain to be further explored. It would be very interesting if the decline in regenerative ability of the heart with age is in zebrafish, in contrast to mammals, not associated with increased replication stress.

*In Figure 4B-F, the percentages of γ H2a.x+ CMs over border CMs in wild-type have huge variation, especially in Fig.4E the average is less than 4 which is much lower than the other panels and Fig.1E-G (around 10-20). The percentages in *bmp2b* OE, *bmp7b* OE and *bmp7a*^{-/-} are actually similar (around 5), the conclusion that the percentage of γ H2a.x+ CMs increases in *bmp7a*^{-/-} is merely due to the low percentage in wild-type in Fig.4E. This holds true for Figure 6C, D, the percentage increases in *bmp7a*^{-/-} is merely due to the low percentage in wild-type in Fig.6D.*

We appreciate the concern of the reviewer. Indeed, it is our observation that the absolute numbers of γ H2a.x+ CMs are not consistent across experiments. We suspect that the experiment-to-experiment variation of γ H2a.x levels reflects multiple variables. First, we have noted that fresh batches of antibodies produce better signal-to noise immunostainings, which allow us to count more CMs as being clearly positive. We have resorted to use HU treated hearts as positive controls for every batch of antibody to ascertain that it remains functional. Still, since we cannot order fresh antibody for every experiment, some variations are due to stainings with batches of antibody of

different age. Secondly, we have shown that γ H2a.x levels correlate with the rate of CM cycling (except under conditions where we manipulate BMP signaling). We and others have repeatedly observed that the fraction of CMs entering the cell cycle at the wound border can vary substantially between groups of fish, despite seemingly identical injury conditions. One major factor explaining these variations appears to be the genetic background. Our collaborator Mathilda Mommersteeg has kindly provided unpublished data showing that different wild type strains respond differently with respect to PCNA levels upon injury (**Figure B in this response to reviewer's**).

Figure B. The fraction of cycling CMs (detected by PCNA immunofluorescence) varies considerably in different wild-type strains at 7 and 21 dpi after cryoinjury.

Since we work with several transgenic lines, which were provided to us in different genetic backgrounds, our experiments were performed in fish of varying backgrounds. Thirdly, some of the variations in the fraction of γ H2a.x+ CMs between experiments can be attributed to additional stress we had to inflict on the fish in some but not other experiments. Some experiments require incubation in drugs without running water. Additionally, for some experiments we have utilized daily injection of EdU. Since we administer 3 such injections intraperitoneally, it adds to the stress on the fish. It is well known that stress can alter the regenerative response and thus also the CM cycling rates (Sallin & Jazwinska 2016, Open Biol, doi: 10.1098/rsob.160012). Of course, we aim to keep stress at a minimum and at levels that do not perturb regeneration, but it is a likely a factor influencing experiment-to-experiment variations.

While we thus cannot avoid differences in CM cycling and replication stress between experiments, we take extreme care to make sure that our controls and experimental fish WITHIN each experiment are as similar as possible. We only use controls (e.g. wild-types) that are siblings of the experimental fish (and thus will share the same genetic background) and raise the fish together, such that any environmental influences apply to all of them. We genotype fish only very shortly prior to the experiment or perform the experiments on unidentified fish to make sure that their feeding status, stocking density etc. is again as similar as possible. We take care to keep control and experimental fish at defined and identical conditions after cryoinjury (during regeneration), in particular always at identical stocking density. Finally, we make sure that control and experimental fish are always treated in identical manner when it comes to potential additional experimental stress (heat-shock, EdU injection, drug treatment etc.).

Thus, despite the issue of experiment-to-experiment variations, we are confident that the results that we report based on comparison of fish within experiments are sound.

The authors showed zebrafish CMs experience replication stress during regeneration but not physiological heart growth, and suggested that the ability to efficiently overcome replication stress is a key reason for the elevated capacity of adult zebrafish to regenerate the heart. When it comes to mouse experiments, people are eager to know if this holds true in neonatal mouse heart regeneration and if this may contribute to the differences in regenerative capability of neonatal and adult mouse hearts.

We agree with the reviewer that it would be very interesting to know to which extent replication stress occurs during neonatal mouse heart regeneration and whether BMP signaling plays a role in alleviating it. Unfortunately, the experiments we have performed to answer these questions have been inconclusive. In collaboration with Mona Malek Mohammadi (University of Bonn) we have first tried to assess by immunofluorescence against γ H2a.x whether neonatal mouse cardiomyocytes display signs of DNA damage/replication stress. Unfortunately, we were only able to obtain non-specific staining in all CM nuclei. We thus resorted to Western blots for γ H2a.x. To avoid the issue that wounded/infarcted tissue might contain other cell types with damaged DNA, we concentrated our analysis on the non-injured right ventricle, which displays robust upregulation of CM cell cycling in mice where the left ventricle was infarcted (see **Figure C** in this response to reviewer's from Hu et al., JCI Insight. 2024;9(5):e176281).

Since the levels of additional CM cycling in the right ventricle induced by infarction of the left ventricle are actually similar to the levels in the left ventricle, if not higher, we assume that replication stress should be detectable there as well, if it poses a challenge to CM cycling in the neonatal mouse heart. We performed Western blots for γ H2a.x and the nuclear marker H3 on right ventricles of mice that underwent sham injury or myocardial infarction and on groups that were treated with the BMP signaling inhibitor LDN193189 (**Figure D in this response to reviewer's**).

As a positive control for our ability to detect DNA damage with this antibody in mouse cells, we also blotted irradiated mouse B-lymphocyte (CLL) cells (**Figure D in this response to reviewer's**). While the positive controls worked, we had a hard time convincing ourselves that the weak and fuzzy bands detected in any of the mouse hearts truly reflect γ H2a.x levels (**Figure E in this response to reviewer's**).

When we nevertheless tried to amplify the signals, we could not detect differences between the four groups. We also performed western blots for the replication stress marker p-Rpa32 and were unable to detect any bands. Unfortunately, we are not able to perform an *in vivo* positive-control for γ H2a.x and p-Rpa32 accumulation like hydroxyurea treatment of mice due to the lack of a corresponding

Figure C. Robust induction of CM cycling in the right ventricle upon myocardial infarction of the left ventricle of neonatal mice. From Hu et al., JCI Insight. 2024;9(5):e176281

Figure D. Western blots for γ H2a.x and nuclear H3 using non-irradiated and irradiated mouse B-lymphocyte CLL line.

animal license. Thus, these results remain inconclusive. We hope that the reviewer agrees that clarifying whether CM replication stress exists in regenerating neonatal mouse hearts is beyond the scope of this manuscript, since it requires additional animal licenses that are very time-consuming to obtain in Germany.

Figure E. Western blots for γ H2a.x and the nuclear marker H3 on right ventricles of neonatal mouse (P1) after sham injury (sham) or myocardial infarction (MI) of the left ventricle. They were either injected with water (solvent) or with the BMP signaling inhibitor

We attempted to gain evidence for or against the existence of replication stress by checking whether upregulation of DNA damage related genes during neonatal mouse heart regeneration can be observed on an existing dataset (GSE123863) from sham injury vs infarction (Wang Z, et al., 2019 <https://doi.org/10.1073/pnas.1905824116>). Upon analyzing gene ontology for differentially expressed genes we could not detect a significant enrichment for DNA damage. Curiously, as in zebrafish we could not find enrichment for DNA replication either. Thus, since we cannot clearly show whether replication stress exists in the regenerative stage of injured neonatal mouse hearts, we can also not answer whether any changes in DNA damage response pathway activities are associated with the loss of regenerative potential.

However, it is worth noting that we have previously published that the levels of BMP7 decrease with neonatal heart growth in mouse (Bongiovanni et al. 2024, Cell Rep, doi: 10.1016/j.celrep.2024.114162). BMP7 stimulates CM proliferation *in-vitro* and *in-vivo* after mouse neonatal heart injury and adult myocardial infarction and its downregulation is thus clearly correlated with the loss of regenerative potential shortly after birth. In this manuscript we show that Bmp7 is an essential regulator of replication stress in the zebrafish heart as well, and that the role of BMP signaling in alleviation of replication stress is well conserved between zebrafish and mammals. Thus, it is tempting to speculate that BMP7 promotes neonatal mouse heart regeneration at least in part also because it does alleviate replication stress.

Other minor concerns:

In Figure 1A, the transcriptomic analysis is based on resection model while the rest of Figure 1 are cryoinjury model. The author could analyze RNA-seq dataset of cryoinjury from database and compare different time points.

We have looked for other datasets where cryoinjury was performed including the meta-analysis of several datasets utilizing cryoinjury (doi: 10.1038/s41598-023-32272-6 and data from the Mommersteeg lab, Oxford university, unpublished). Unfortunately, we do not see a consistent signature of genes annotated with GO terms related to DNA repair in these datasets. However, amazingly we also do not see enrichment of GO terms related to DNA replication. Clearly cryoinjury does robustly induce CM cell cycling, thus these datasets seem to lack the sensitivity to detect this at

the gene expression level. The above-mentioned published manuscript describes cryoinjury response as noisy with inherent variability across laboratories and data also varies with the strain of fish used. Therefore, we suspect the generally noisy nature of cryoinjury to mask some processes as critical as DNA replication and repair. While we are unable to confirm our transcriptomic data derived from resected hearts with other published transcriptomics from cryoinjured hearts, it is worth pointing out that we have confirmed all our initial conclusions derived from the transcriptomics experimentally in cryoinjured hearts, including the gene expression data of DNA repair-associated genes by qPCR in Figure 1B.

Figure 2A, the author should also provide the percentage of γ H2a.x+ and γ H2a.x- in EdU+ CMs.

To address this, we have added a new pie chart to **Figure 2A**. The majority (55%) of EdU+ CMs are also γ H2a.x+, demonstrating that replication stress does not only affect a minority of cycling CMs. We describe these new results on the lines 115-119 as follows:

“We found that ~55% of the EdU+ CMs were also positive for γ H2a.x. Conversely, 81% of the γ H2a.x+ CMs were EdU-positive, showing that γ H2a.x+ was largely confined to CMs that had recently been cycling or were still in a cycling state. These data suggest that CMs become γ H2a.x positive because they experience replication stress when they enter the cell cycle during regeneration.”

Figure 2C, the main text stated unperturbed regenerating hearts but in the diagram indicated HU treatment.

We thank the reviewer for pointing this out. We have corrected the cartoon.

*Do *bmp7a* and *bmp7b* have similar functions? The overexpression is using *bmp7b* while the knockout is *bmp7a*.*

We do not think that there is a difference between effects of *bmp7a* and *bmp7b* in overexpression assays. We have confirmed the same in a previous publication where overexpression of *bmp7a* or *bmp7b* RNA had the same developmental effects in zebrafish (Bongiovanni et al., Cell Rep. 2024, doi: 10.1016/j.celrep.2024.114162). The endogenous *bmp7a* gene is more strongly expressed during regeneration, thus we studied the *bmp7a* mutant. Yet, for overexpression assays it should not matter whether *bmp7a* or *bmp7b* are used, thus we used an available line for *bmp7b* overexpression.

The title may be a little overstated, the current data only suggest BMP signaling promotes ZEBRAFISH heart regeneration via alleviation of replication stress.

We have modified the title. It now reads "BMP signaling promotes zebrafish heart regeneration via alleviation of replication stress".